# Genome-wide dynamics of Pol II elongation and its interplay with promoter proximal pausing, chromatin, and exons

**Iris Jonkers[1]\*[†], Hojoong Kwak[2†], John T Lis[1]\***

[1]Department of Molecular Biology and Genetics, Cornell University, Ithaca, United States; [2]Howard Hughes Medical Institute, University of Michigan, Ann Harbor, United States

**Abstract** Production of mRNA depends critically on the rate of RNA polymerase II (Pol II) elongation. To dissect Pol II dynamics in mouse ES cells, we inhibited Pol II transcription at either initiation or promoter-proximal pause escape with Triptolide or Flavopiridol, and tracked Pol II kinetically using GRO-seq. Both inhibitors block transcription of more than 95% of genes, showing that pause escape, like initiation, is a ubiquitous and crucial step within the transcription cycle. Moreover, paused Pol II is relatively stable, as evidenced from half-life measurements at ~3200 genes. Finally, tracking the progression of Pol II after drug treatment establishes Pol II elongation rates at over 1000 genes. Notably, Pol II accelerates dramatically while transcribing through genes, but slows at exons. Furthermore, intergenic variance in elongation rates is substantial, and is influenced by a positive effect of H3K79me2 and negative effects of exon density and CG content within genes.

**\*For correspondence:**
ihj3@cornell.edu (IJ);
johnlis@cornell.edu (JTL)

[†]These authors contributed equally to this work

**Competing interests:** The authors declare that no competing interests exist.

**Reviewing editor**: Kevin Struhl, Harvard Medical School, United States

## Introduction

Many steps throughout the transcription cycle of RNA polymerase II (Pol II) can be regulated, and modulation of any step has the potential to alter the timing and output of mRNA production. After initiation of Pol II, transcription regulation is mediated mostly by the dynamics of Pol II elongation. For example, +20 to 100 nts downstream of the transcription start site (TSS), Pol II can be slowed down and paused by negative elongation factor (NELF), DRB-sensitivity inducing factor (DSIF) and core promoter components (*Adelman and Lis, 2012*; *Kwak et al., 2013*). The escape of Pol II from the paused state into productive elongation can be rate-limiting, and is dependent on the positive elongation factor P-TEFb, which consists of the Cdk9 kinase and CyclinT1 (*Marshall et al., 1996*; *Lis et al., 2000*; *Ni et al., 2008*). P-TEFb is recruited directly or indirectly to the paused Pol II complex by transcription activators, where it phosphorylates the C-Terminal domain (CTD), as well as DSIF and NELF, transforming DSIF into a positive elongation factor and evicting NELF (*Peterlin and Price, 2006*). P-TEFb appears to be both necessary and sufficient for paused Pol II escape into productive elongation; blocking P-TEFb kinase activity with the drug flavopiridol (FP) (*Chao and Price, 2001*) in *Drosophila* causes an increase of promoter proximal Pol II at the majority of genes (*Henriques et al., 2013*). The accumulation of paused Pol II has been proposed to be the result of rapid rounds of termination and re-initiation, creating a highly dynamic Pol II peak at the promoter proximal region (*Brannan et al., 2012*; *Davidson et al., 2012*). However, paused Pol II in *Drosophila* seems remarkably stable as shown by extensive kinetic and in vivo analysis at the *Drosophila Hsp70* locus (*Buckley et al., 2014*) and by estimation of decay rates of over a dozen *Drosophila* genes by blocking TFIIH helicase activity, and thereby initiation, with the drug triptolide (Trp) (*Henriques et al., 2013*). Thus, *Drosophila*

**eLife digest** Many different factors determine how quickly the DNA in genes is transcribed to produce molecules of messenger RNA. The start of the transcription process features two milestones: first, an enzyme called RNA Polymerase II starts the process; shortly afterwards, however, the process pauses and only starts again when other proteins are recruited. This provides two levels of control over the production of messenger RNA and, it also allows the transcription process to be interrupted in order to study the rate of transcription.

Here, Jonkers, Kwak and Lis used two drugs to block either the start of transcription or the release from the paused state in mouse cells. Both drugs prevented new transcription and disrupted about 95% of the total number of genes. However, RNA Polymerase II that was already copying DNA could continue to copy, and did so at an average rate of 2000 bases per minute. Transcription rates were, however, shown to vary between different genes—highly active genes are transcribed faster. Transcription rates also varied within individual genes, with the enzyme accelerating as it moves along the gene. This suggests that the transcription machinery, including other proteins that improve the enzyme's efficiency, are recruited or modified after transcription has already started, and that these proteins help the enzyme to reach its maximum transcription speed.

Other factors also affected the transcription rate: the genetic code is written in four letters—A, C, G and T—and genes that contained more Cs and Gs were transcribed slower than those with lots of As and Ts. Genes also contain regions called exons that code for proteins, and regions called introns that do not: Jonkers, Kwak and Lis found that genes with lots of exons were transcribed slower. Furthermore, DNA is wrapped around proteins into a compacted structure, and genes that had certain chemical markings added to these proteins were transcribed faster.

The work of Jonkers, Kwak and Lis is the first in-depth look at how transcription is affected by gene structure, and leads the way to uncovering how transcription rates throughout genes are regulated to influence production of messenger RNA.

Pol II transcription can be regulated by the promoter proximal, stable pausing and by transcription factor-controlled entry of paused Pol II into productive elongation in *Drosophila*.

In mammals, promoter proximal pausing also seems to be a P-TEFb dependent and rate-limiting step during early elongation for many genes (*Core et al., 2008*; *Rahl et al., 2010*). Expressed genes without a peak of paused Pol II in one cell type, may acquire pausing in another (*Min et al., 2011*), indicating that genes have the potential of becoming regulated by promoter proximal pausing even when a promoter proximal Pol II peak is absent. However, it is unclear if all genes undergo this P-TEFb dependent step in mammals, and how stable the paused Pol II is.

Downstream of the promoter proximal region, the rate of Pol II elongation has been proposed to influence (co-)transcriptional processes such as splicing (*Howe et al., 2003*; *de la Mata et al., 2003*; *Shukla and Oberdoerffer, 2012*), 3′ end processing (*Nag et al., 2007*), termination (*Hazelbaker et al., 2012*), and overall levels of mRNA (*Danko et al., 2013*). Genic features that could potentially influence elongation rates are often, but not exclusively, associated with exons. Examples are histone modifications around exons and within the gene body (*Kolasinska-Zwierz et al., 2009*; *Schor et al., 2009*; *Schwartz et al., 2009*; *Kim et al., 2011*; *Saint-André et al., 2011*; *Schor et al., 2013*), nucleosome occupancy (*Schwartz et al., 2009*; *Tilgner et al., 2009*), GC content of both intronic and exonic DNA (*Schwartz et al., 2009*; *Amit et al., 2012*), and DNA methylation (*Maunakea et al., 2013*). Many exonic chromatin features have been implicated in regulation of splicing as well (*Kolasinska-Zwierz et al., 2009*; *Schor et al., 2009*; *Shukla et al., 2011*). Moreover, Pol II has been shown to accumulate preferentially at spliced exons (*Brodsky et al., 2005*; *Alexander et al., 2010*; *Chodavarapu et al., 2010*; *Kwak et al., 2013*), leading to the hypothesis that reduction of Pol II elongation rate facilitates splice site recognition and spliceosome assembly, and thus, splicing of the associated intron (*Shukla and Oberdoerffer, 2012*). Indeed, transcription with a slow mutant of Pol II promotes alternative splicing in human and yeast cells (*Howe et al., 2003*; *de la Mata et al., 2003*; *de la Mata et al., 2010*).

The relationship of elongation rates, exons and other features of transcription units is still in dispute. Despite observations of Pol II accumulation at exons, three studies that directly measured elongation rates at multiple genes could not clearly demonstrate a correlation between exons and

elongation rate (*Singh and Padgett, 2009*; *Brody et al., 2011*; *Danko et al., 2013*). Furthermore, although in vitro studies clearly show the effect of nucleosomes and histone modifications on elongation rate (*Orphanides et al., 1998*; *Hodges et al., 2009*; *Bintu et al., 2012*), the in vivo consequences of chromatin on elongation rate are less understood, primarily because of the inability to measure elongation rates at many genes simultaneously. Previous studies have measured elongation rates ranging from 1 to 4 kb/min at individual genes in various organisms (*Ardehali and Lis, 2009*). Recently, elongation rates for over 160 genes were measured simultaneously by following the induction wave of Pol II after estradiol or TNF-alpha treatment (*Danko et al., 2013*). Interestingly, this study showed a broad range of elongation rates between and within cell types, suggesting that the control of elongation rate may be used to regulate transcription and co-transcriptional processes. Thus far, elongation rates have only been measured in rapidly inducible genes, which limits the analytical power to reveal the associations to various features of transcription. Therefore, it is critical to expand the number of elongation rates measured simultaneously in vivo, to allow systematic analysis of the correlation between exons, chromatin and elongation rate.

In this study, we use Trp and FP, two highly specific drugs to block initiation or pause escape, in combination with the sensitive GRO-seq assay (*Core et al., 2008*) to examine the drug-induced kinetic changes in Pol II distribution over promoter proximal regions and in the gene body. While FP blocks escape of paused Pol II, elongating Pol II can still clear the gene, and both changes can easily be followed by the sensitive and transcription orientation specific GRO-seq assay. We definitively show that P-TEFb is required for paused Pol II to escape into a productive elongation, providing a platform for transcription regulatory input on the early elongation rate, even for genes not previously known to be paused, confirming and extending earlier results using ChIP-seq of Pol II (*Rahl et al., 2010*). Similarly, use of Trp to block Pol II initiation and entry to the pause, allows kinetic analysis of paused Pol II stability on nearly 3200 genes, showing that paused Pol II has a relatively long half-life and excluding rapid termination mechanisms as a major factor of Pol II regulation at the promoter. Furthermore, inhibition of Pol II gene body entry causes a 'wave' of elongating Pol II that, when assayed as a function of time after FP or Trp addition, allows measurement of elongation rates at over 1000 genes simultaneously, and over different regions within genes. We show that Pol II elongation rates increase within the gene body, suggesting a gradual maturation of the elongation complex as it progresses through the gene. In addition, we analyzed elongation rate variation as a function of a large number of genic and chromatin characteristics. Elongation rates correlate negatively with exon density and CpG content and methylation, and positively with active transcription mark H3K79me2. Overall, we can explain ~30% of the gene-by-gene variation in elongation rates. Our study of the dynamics of Pol II shows that elongation rate is highly dynamic at all genes, both at the promoter proximal region and within the gene body.

## Results

### Generation of GRO-seq libraries treated with FP and Trp

When a gene is activated, P-TEFb kinase is recruited to promoters and phosphorylates the paused Pol II·NELF·DISF complex, allowing paused Pol II to more rapidly escape into productive elongation. To identify all genes dependent on P-TEFb, we inhibited P-TEFb kinase activity with the drug FP. For comparison, we also inhibited pre-initiation complex formation with Trp. We isolated replicates of nuclei from untreated mESCs and cells treated for 2, 5, 12.5, 25 and 50 min with 300 nM FP, as well as nuclei treated for 12.5, 25, and 50 min with 500 nM Trp (*Figure 1A*) and performed GRO-seq with these nuclei (*Figure 1—source data 1*). To minimize off-target and secondary effects, we first determined the minimum concentrations of FP and Trp required to clear Pol II from the *ActB* gene body using ChIP-qPCR with an antibody to total Pol II (not shown), and then used these concentrations, which were at the lower spectrum compared to previous studies (*Chao and Price, 2001*; *Rahl et al., 2010*; *Titov et al., 2011*), in our genome-wide analyses. Furthermore, we ensured that drug treated mESCs were morphologically indistinguishable from untreated cells. Biological replicates correlated extremely well (*Figure 1—source data 2 and 3*) and were combined for further analysis. Because inhibition of P-TEFb and initiation were anticipated to have large genome-wide effects on Pol II transcription, we normalized treated and control libraries to in vitro transcribed *Arabidopsis thaliana* RNAs added during the run-on.

To assess the cellular effects of the drugs on Pol II and the phosphorylation of its CTD, we fractionated insoluble chromatin from the soluble cytoplasmic/nucleoplasmic fractions of control and drug

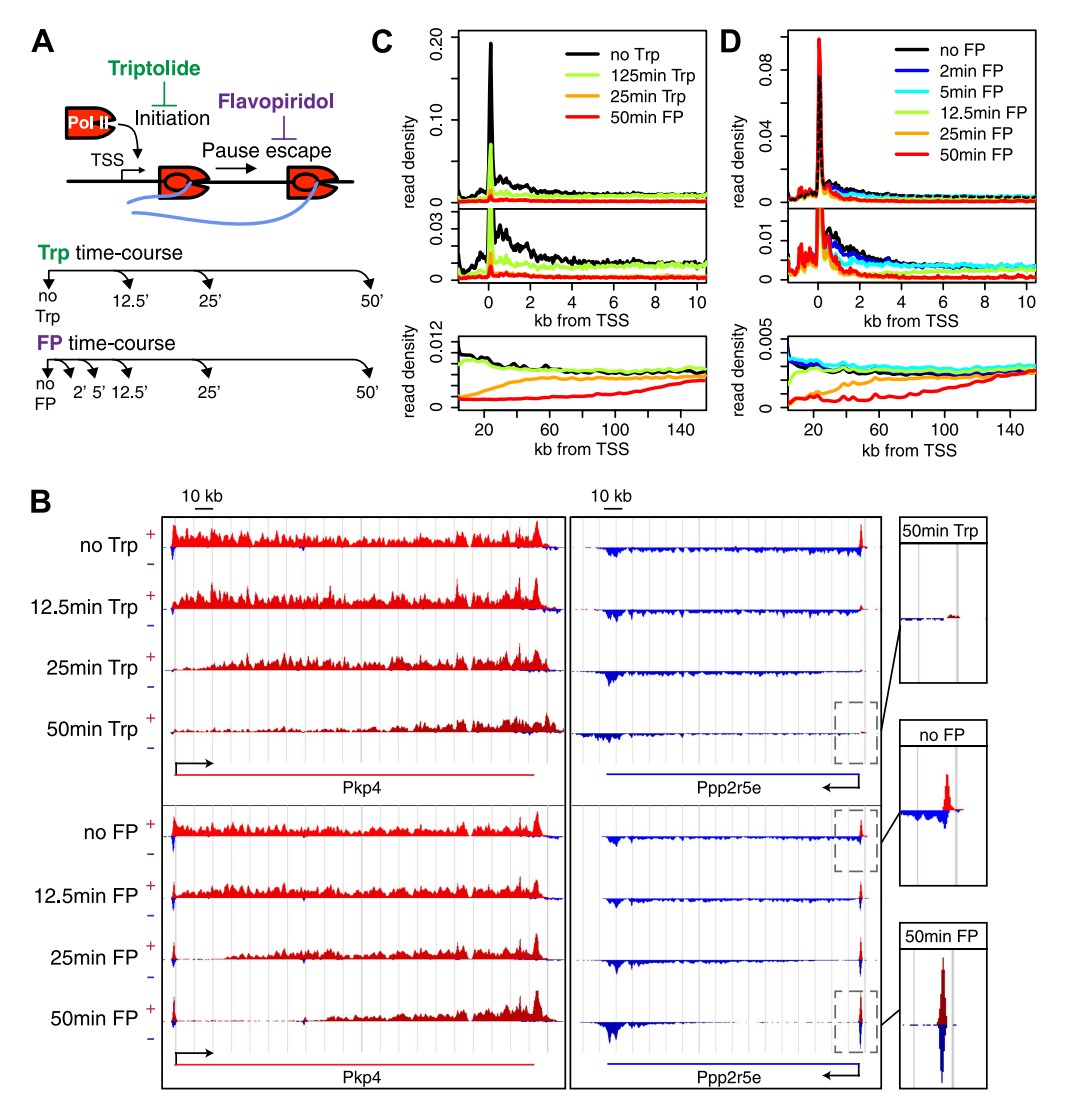

**Figure 1**. Timed inhibition of pause escape (P-TEFb) or initiation (TFIIH) has similar effects on the gene body Pol II distribution, but opposite effect at the promoter-proximal region. (**A**) Experimental set-up, 300 nM flavopiridol (FP) and 500 nM of triptolide (Trp) were used to block pause escape or transcription initiation in mES cells. Nuclei were isolated for GRO-seq at timepoints after treatment as specified. (**B**) Screenshot of genes Pkp4 and Ppp2r5e with or without Trp or FP treatment for 12.5, 25 or 50 min, with sense reads in red and antisense reads in blue. (**C**) Composite profile of GRO-seq read density of all genes >12.5 kb (top panel) or >150 kb (bottom panel) after treatment with Trp for various durations of time. The middle panel is a zoom-in of the top panel. The bottom panel shows the region downstream of the TSS for genes longer than 150 kb. (**D**) As (**C**), but after timed treatment with FP.

The following source data and figure supplements are available for figure 1:

**Source data 1**. Sequencing and alignment of GRO-seq replicates in the Trp and FP time courses.

**Source data 2**. Pearson correlation of GRO-seq replicates in the Trp and FP time courses.

**Source data 3**. Spearman correlation of GRO-seq replicates in the Trp and FP time courses.

**Figure supplement 1**. Inhibition of the P-TEFb kinase by Flavopiridol.

**Figure supplement 2**. Stability of Pol II elongation after the inhibition of initiation or pause escape.

treated mESCs and performed western blots with antibodies against the N-terminus of Rpb1, and the Serine5 or Serine2 phosphorylated CTD (*Figure 1—figure supplement 1A*). Chromatin bound Pol II is reduced after treatment with either FP or Trp (*Figure 1—figure supplement 1B*). However, phosphorylation of the CTD of chromatin bound Pol II was reduced only after FP treatment, but not Trp (*Figure 1—figure supplement 1C*). Overall, these results indicate that FP and Trp exert the intended effects on the phosphorylation of the Pol II CTD.

Two long genes, Pkp4 and Ppp2r5e (*Figure 1B*), and composite profiles of all genes (*Figure 1C,D*) illustrate the effects of P-TEFb inhibition by FP (pause escape) and TFIIH inhibition by Trp (initiation) on Pol II distribution. Both drugs work rapidly, as gene body density near the TSS approaches background levels within 12.5 min. The rapid drug action and immediate measurements thereafter minimize the possibility of major secondary effects within the short timeframe of the experiment (*Figure 1C,D*). Furthermore, as the FP and Trp dependent block of Pol II entry into the gene persists, elongating Pol II forms 'inhibition waves' that are very similar between treatments (*Figure 1C,D*, lower panels). Downstream of these inhibition waves, Pol II density levels remain equal throughout the time course, indicating that elongating Pol II in the gene body is not affected (*Figure 1—figure supplement 2A,B*). Conversely, the two drugs have opposite effects at the promoter proximal region. The composite paused Pol II peak increases after FP treatment (*Figure 1B,D*), but disappears after Trp treatment (*Figure 1B,C*). These results confirm that Trp blocks transcription initiation and causes the time dependent clearing of Pol II from the promoter and gene body, while FP's prominent effect is to block escape from the pause, causing a time dependent gene body clearance but enhanced promoter proximal Pol II pausing.

## P-TEFb dependent escape from the pause occurs at all active TSSs

Recently, *Henriques et al. (2013)* observed in *Drosophila* that the majority of genes are susceptible to P-TEFb inhibition. To extend this result to mammals, we quantified changes in Pol II distribution near the TSS after FP treatment and compared it to the effects of Trp treatment. We selected the top 75% actively transcribed genes that are over 3.5 kb in length and well resolved (n = 6380, *Figure 2—figure supplement 1A*), and generated heat-maps of the Pol II density ± 2 kb around the TSSs of each gene before and after 50 min of drug treatment on the sense or antisense strand (*Figure 2*, *Figure 2—figure supplement 1B*). The antisense strand shows the presence of upstream divergent transcription, a well-documented feature of mammalian promoters (*Core et al., 2008*; *Seila et al., 2009*; *Flynn et al., 2011*). While Trp causes a reduction in the promoter and downstream regions of the annotated gene (sense strand), and the upstream divergent region (antisense strand) (*Figure 2A*, *Figure 2—figure supplement 1B*), FP inhibition causes the opposite effect on the promoter regions (*Figure 2B*, *Figure 2—figure supplement 1B*). Pol II density in the gene body is decreased after FP treatment, but Pol II increases in promoter proximal and divergent regions at the majority of genes (*Figure 2B*, *Figure 2—figure supplement 1B*; *Figure 2—source data 1*). These results indicate that FP generally allows newly recruited Pol II to enter into early elongation and pausing, but blocks the entry into productive elongation. While a smaller fraction (~20%) of genes displayed an unexpected decrease in paused and divergent Pol II peaks (*Figure 2—source data 1*; *Figure 2B*), these genes generally display lower levels of pausing and gene body transcription (*Figure 2B*). This suggests that FP can reduce initiation or increase termination of weakly paused and less active genes, either directly or indirectly. Generally, the change in Pol II promoter density after FP inhibition correlates positively with the preexisting level of pausing and productive transcription of a gene (*Figure 2C*), suggesting that efficient initiation is a prerequisite for an FP-induced increase in paused Pol II.

Quantification of the decrease in read density directly downstream of the TSS shows that 96% of genes are significantly decreased after Trp treatment, while 95% of genes are decreased after FP treatment (*Figure 2—source data 1*). This demonstrates that P-TEFb-dependent Pol II entry into productive elongation is as universal a step in gene transcription as TFIIH helicase-dependent initiation.

## Paused Pol II is relatively stable

After blocking Pol II initiation with Trp, the level of promoter proximal Pol II will decrease in time with a rate comprising both escape and termination. This decay rate indicates the stability of paused Pol II, from which we can infer whether pausing is regulated by continuous cycling of termination and reinitiation, characterized by a high decay rate and a minimal half-life of paused Pol II, or whether paused Pol II is relatively stable, and is released into productive elongation by a positive signal like recruitment and phosphorylation by P-TEFb. We calculated decay rates of Pol II after Trp treatment using a first

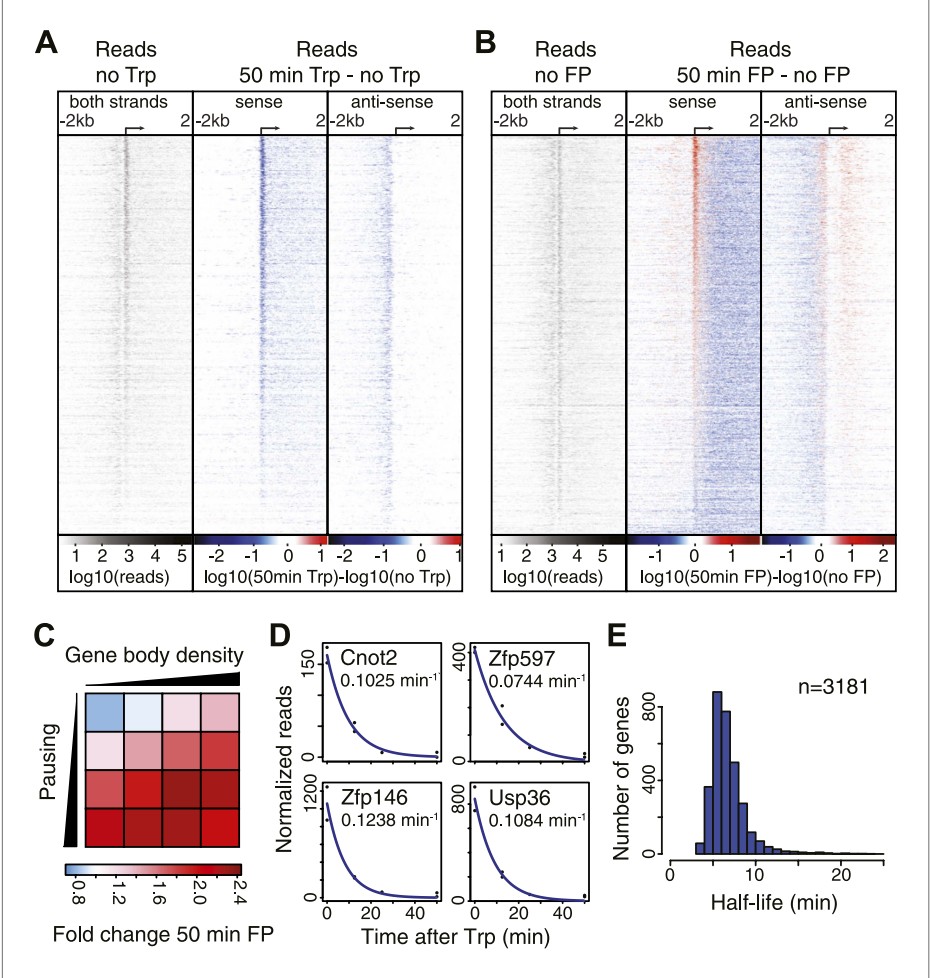

**Figure 2**. Pol II distribution and change at the TSS region of active genes after pause escape (P-TEFb) or initiation (TFIIH) inhibition. (**A**) Left panel shows a density plot of the $log_{10}$ of reads of the no Trp dataset, in 10 bp windows ±2 kb around TSSs of active genes. The two right panels show the $log_{10}$ difference in 10 bp windows between 50 min Trp treatment and no treatment for the sense strand (middle) and antisense strand (right panel) with decrease in blue and increase in red. Genes are ordered by the maximum decrease after Trp treatment at the promoter-proximal region. The density scales and color keys for each panel are depicted the bottom. (**B**) As in (**A**), $log_{10}$ reads in 10 bp windows around the TSS of the no FP control dataset (left panel), and the $log_{10}$ difference in reads between 50 min FP treatment and no treatment for the sense (middle panel) and antisense strand (right panel). Genes are ordered by maximum increase after FP treatment at the promoter-proximal region. (**C**) The average fold change of promoter proximal Pol II in a 250 bp region in the sense direction after 50 min FP treatment in quartiles of pausing, as measured by the pausing index (y-axis), and activity, as measured by GRO-seq gene body density (x-axis). (**D**) Paused Pol II decay rates are calculated by non-linear regression with an exponential decay model over the read counts in the promoter-proximal region of each replicate time point in the Trp time course for each individual gene. Four examples of decay rate measurement with the standard deviation and the regression are shown, with the reads per time point per replicate shown as black dots, and the blue line as the regression of the decay. (**E**) The half-life's of Pol II in promoter proximal regions of 3181 genes that have a high confidence decay rate with standard deviations (<0.5 times the decay rate).

The following source data and figure supplements are available for figure 2:

**Source data 1**. Change in promoter proximal and gene body read density, and pausing index after treatment with FP or Trp in time.

**Figure supplement 1**. Differential dynamics of paused Pol II upon the inhibitions of initiation or the pause escape.

order exponential decay model, and found high confidence decay rates for 3181 genes (standard deviation <0.5 times the decay rate) (*Figure 2D,E*). The mean half-life of Pol II is 6.9 min, in agreement with the decay rates measured in *Drosophila* (*Henriques et al., 2013*; *Buckley et al., 2014*). Thus, promoter proximal Pol II stability is comparable between species and paused Pol II is relatively stable.

## Measuring the speed of elongating Pol II in the gene body

Pol II that is already transcribing when FP is added generates a clearly distinguishable wave as time progresses. We tracked the rate of this wave's progress in an unbiased manner with a hidden Markov model (HMM, *Figure 3—figure supplement 1A*) at more than 1000 long and actively transcribed genes (*Figure 3A*). The HMM was applied to each biological replicate, and only genes with reproducible results were used. We also tracked the wave following Trp addition as an independent strategy for blocking Pol II entry into genes (*Figure 3—figure supplement 1B,C*). The Trp inhibition wave was less clearly defined than the FP wave, likely due to the fact that Pol II initiated and paused before Trp treatment will gradually escape after TFIIH inhibition, resulting in a less defined block of Pol II entry into the gene body (*Figure 3A*, *Figure 3—figure supplement 1B*, see Med13l). The middle of the inhibition wave identified by the HMM was defined as the transition point between the affected and unaffected region of the gene body.

On average, Pol II travels about 100 kb during 50 min of FP treatment. This corresponds to an elongation rate of about 2 kb/min, which is comparable to Pol II rates from previous studies (*Ardehali and Lis, 2009*; *Danko et al., 2013*). The average Pol II inhibition wave travels at identical rates after either Trp or FP treatment, indicating that inhibition of P-TEFb or off-target effects of FP are unlikely to have significant effects on elongation rates downstream of the promoter proximal region (*Figure 3B*). Although the HMM did not always derive transition points for each timepoint of drug treatment within genes, the average transition points of genes with detectable inhibition waves at three time-points behaved identically to the average of the genes where the HMM derived only one or two transition points (*Figure 3—figure supplement 1D*). Similarly, for genes where both FP and Trp transition points were reliably measured, the average FP and Trp transition points aligned well (*Figure 3—figure supplement 1E*). Both observations indicate that the measurements of transition points are unaffected by the groups of genes for which we measured the transition points after either treatment, or the method of inhibition.

Next, we calculated elongation rates for genes that had two or more consecutive transition points. Since we do not include 0 min as a baseline, and start measuring transition points after 5 min when significant changes in Pol II distribution are observed, the lag time of the inhibitor action does not affect the elongation rate estimations. We directly measured 141 early (5–12.5 min), 938 mid (12.5–25 min), and 245 late (25-50 min) elongation rates, with average elongation rates of 0.5, 1.8 and 2.4 kb/min respectively (*Figure 3—source data 1*). The variation within each group of elongation rates was considerable, with standard deviations of 0.43 kb/min, 0.69 kb/min and 0.79 kb/min for the early, mid and late rates, respectively.

Thus, the direct measurement of elongation rates after inhibition of Pol II gene entry by FP or Trp leads to two main observations. First, elongation rates vary considerably between genes, and second, the elongation complex seems to accelerate as it transcribes through the gene. We proceeded to confirm both observations independently.

## Gene-to-gene variation in Pol II elongation rate

To verify our directly measured elongation rates, we compared the mid elongation rates group (n = 938), to an independent measurement of relative elongation rates (*Ameur et al., 2011*), using intronic reads of ribosomal RNA depleted total RNA-seq data from *Sigova et al. (2013)* in mESCs. This method is based on the observation that the majority of intronic RNA is spliced and degraded co-transcriptionally (*Djebali et al., 2012*; *Tilgner et al., 2012*), producing a saw-tooth pattern corresponding to RNA-seq reads in introns across genes (*Figure 3—figure supplement 2A*; *Ameur et al., 2011*). The gradient of RNA density from the 5′ to 3′ splice sites (SS) of the intron, normalized for gene expression by dividing intronic reads with the average read counts near the 3′ end of the introns, reflects the time Pol II spends during elongation along the intron. Slow Pol II will take longer to transcribe an intron, accumulating more RNA-seq reads prior to splicing and degradation, and resulting in a steep intronic RNA-seq read gradient. Fast Pol II will spend less time in an intron, reaching the 3′ SS and subsequent

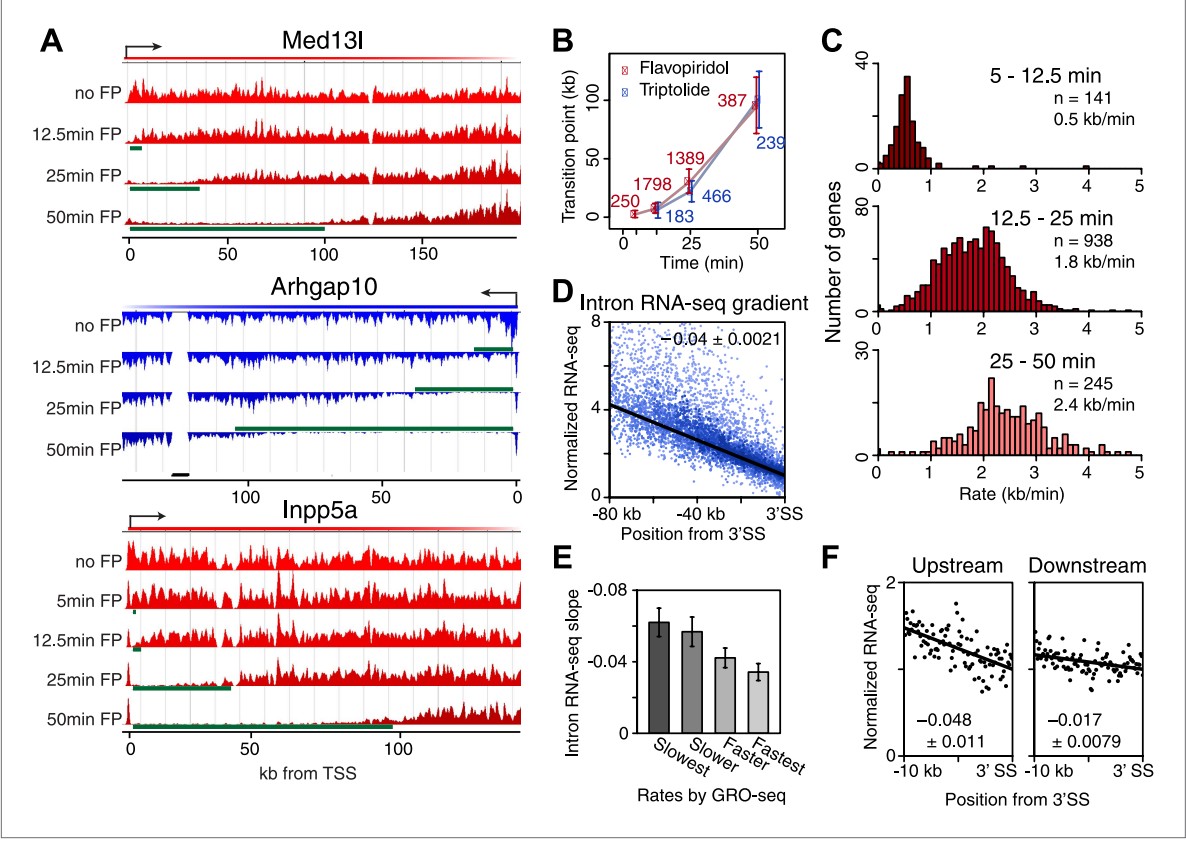

**Figure 3**. Genome-wide determination of Pol II elongation rates after FP and Trp treatment. (**A**) Three representative genes used for measurement of transition points with the HMM. In green, the affected region after FP treatment as established by the HMM, the transition point being the endpoint of this region. In black, repeat regions where reads couldn't be aligned uniquely. (**B**) Average of HMM derived transition points of the FP timecourse (red) or Trp timecourse (blue) plotted against the time of drug treatment. Error bars are standard deviation from the mean. The numbers of the genes with high confidence HMM measurements are next to each data point. (**C**) Elongation rates derived from the FP time course. Elongation rate is the distance traveled in the time spanning 5–12.5 min (top; n = 141), 12.5–25 min (middle; n = 938) and 25–50 min (bottom; n = 245). All elongation rates can be found in the **Figure 3—source data 1**. (**D**) RNA-seq intron gradient relative to the 3′ splicing site (3′SS). Introns longer than 10 kb (n = 17,828) in the refseq gene list are grouped by their sizes, and the average RNA-seq read count per 100 bp bins are plotted by the distance from the 3′SS for each group. Read density in the windows is normalized to the level of reads at the 3′ SS, to compensate for expression differences between genes. The average and the standard deviation of the slope are shown. (**E**) The RNA-seq gradients of the mid elongation rate genes (12.5–25 min) in the introns (1650 introns longer than 10 kb in 938 genes) grouped by the quartiles of the elongation rate (n = 413, 415, 411, 411 respectively for the slowest, slower, faster, and the fastest). Note that the slower genes have greater negative slope than the faster genes. (**F**) The RNA-seq intron gradients of the refseq introns upstream of 25 kb from TSS (n = 380) and introns downstream of 50 kb (n = 854) on the same gene for a smaller region.

The following source data and figure supplements are available for figure 3:

**Source data 1**. Full list of elongation rates derived after FP treatments between times spanning 5–12.5 min, 12.5–25 min, and 25–50 min.

**Figure supplement 1**. Determination of elongation rates using the Hidden Markov Model of inhibition time-points.

**Figure supplement 2**. Determination of elongation rates using the intron RNA-seq gradients.

**Figure supplement 3**. Acceleration of Pol II and its accompanying CTD phosphorylation.

splicing and degradation of intronic RNA faster, resulting in a flatter intronic read pattern and a lower slope (**Figure 3—figure supplement 2B**).

Because the saw tooth pattern was more clearly distinguishable in larger introns and because intronic reads are much less abundant than exonic reads in typical RNA-seq datasets, we grouped introns and used pooled intronic read counts on all introns longer than 10 kb. Similar to what was

shown in an individual gene example (*Figure 3—figure supplement 2A*), we found a negative slope of the read densities as a function of the distance from the 3′ SS (*Figure 3D*). GRO-seq density after 25 min FP treatment in the mid elongation rate genes ordered from slow to fast shows clear alignment of the Pol II inhibition wave (*Figure 3—figure supplement 2C*, left). To verify the observed gene-to-gene variation in elongation rates of these genes, we split the 938 mid elongation rates genes into quartiles. The slope of the intronic density plots of the quartiles decreases as the directly measured elongation rates become faster (*Figure 3E*, *Figure 3—figure supplement 2C*, right), independently confirming the gene-to-gene variation in elongation rates.

## Independent confirmation of the acceleration of Pol II in the gene body

Next, we explored the acceleration of Pol II within the gene body in independent ways: (1) Directly measuring elongation rates in different regions within genes, (2) Analyzing relative elongation rates within different regions of the gene body with intronic RNA-seq read gradients.

First, our direct measurement of early, mid and late elongation rates after either FP or Trp treatment suggests that Pol II accelerates as it transcribes through the gene body (*Figure 3C*, *Figure 3—figure supplement 1C*), which is also evident from the non-linear increase of all the HMM-derived transition points plotted against time (*Figure 3B*, *Figure 3—figure supplement 1D,E*). Moreover, elongation rates derived from multiple regions within the same gene were significantly greater further downstream in the gene body (*Figure 3—figure supplement 3A*), showing that this acceleration occurs within most genes and is not a consequence of elongation rate variation between genes. Also, *Danko et al. (2013)* recently showed a similar acceleration of elongation rates within the gene body for estrogen or TNF-alpha induced genes.

Second, we examined the read density in introns, as described above, in the region from the TSS to 25 kb, and in introns 50 kb downstream of the TSS within the same genes. The slope of intron read density as a function of distance to the 3′ SS is higher in the upstream region compared to the downstream region (*Figure 3F*), indicative of slower elongation rates upstream in genes, and acceleration as Pol II travels within the gene body.

The acceleration is not constant, but decreases as transcription proceeds (*Figure 3—figure supplement 3B,C*), suggesting that the Pol II elongation complex 'matures' and reaches its maximum speed while transcribing a gene. The maturation could be caused by the stochastic accumulation of elongation factors to the Pol II complex, or by gradual stochastic loss of pausing factors. Indeed, pausing negatively correlates with mid elongation rates, while late elongation rates are not affected (*Figure 3—figure supplement 3D*), suggesting that pausing delays the maturation of Pol II. Furthermore, ChIP-seq composite profiles of total Pol II and of Ser5 or Ser2 phosphorylated CTD (*Rahl et al., 2010*) show that Pol II Ser2 phosphorylation, which is a product of P-TEFb activity and other kinases (*Bartkowiak et al., 2010*; *Devaiah et al., 2012*) and presumably coincides with the loss of NELF and DSIF pausing activity, increases gradually within the gene body within the same region as the acceleration of the elongation complex (*Figure 3—figure supplement 3E,F*). This suggests that Pol II is modified gradually while it transcribes, leading to maturation and increased elongation rates.

## Pol II stably elongates in the gene body, while termination is negligible

The GRO-seq density on the gene body shows the amount of productively elongating Pol II and reflects the transcriptional activity of genes. However, the average GRO-seq density shows a decrease as the Pol II travels into the body of the gene (*Figure 1C,D*, top panels). The decreasing pattern of gene body Pol II density in itself could be explained by the gradual termination of Pol II (*Figure 4A*, top; *Figure 4—figure supplement 1B*), but alternatively, an accelerating Pol II could on its own produce a decreasing gene body Pol II density (*Figure 4A*, bottom; *Figure 4—figure supplement 1D*). Thirdly, a mixed model can be envisioned where a decrease in termination and acceleration go hand in hand as Pol II progresses in the gene body (*Figure 4—figure supplement 1C*; *Mason and Struhl, 2005*). The directly measured elongation rates (*Figure 3C*) can be used to assess the degree to which termination and acceleration models contribute to the observed Pol II density decrease. If there is no loss of Pol II through termination, then Pol II density should be inversely proportional to the elongation rate. Therefore, comparing the inverse of the measured elongation rates ($v^{-1}$) with the Pol II density ($D$) can be used to determine which model of Pol II elongation is more probable (*Figure 4A*). On average, the inverse of the elongation rate ($v^{-1}$) does not fall below the density plot ($D$) in the gene body region (*Figure 4B*) where the rates are measured (starting from the average 5 min transition point 2.3 kb

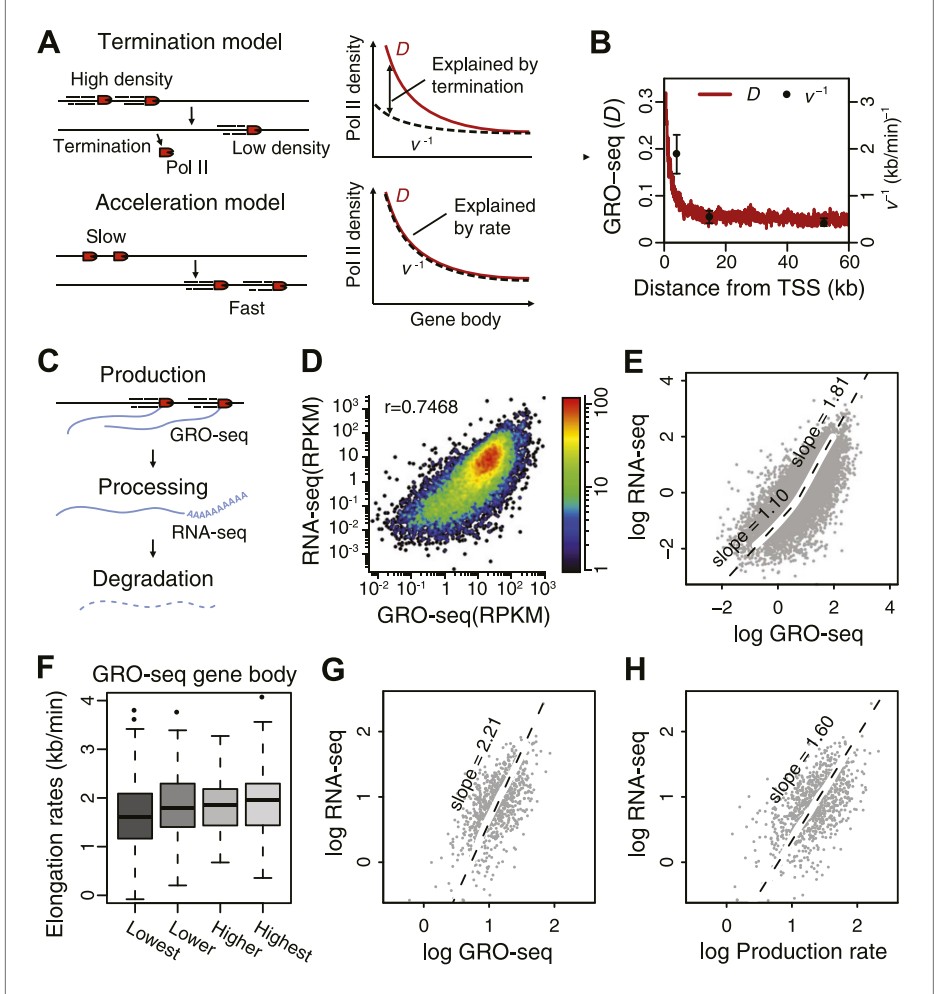

**Figure 4**. Elongation rate explains transcriptional stability and the steady-state output. (**A**) Models of Pol II elongation (left), and the expected steady-state GRO-seq density (*D*) and the inverse elongation rates ($v^{-1}$) for each model (right). The termination model proposes that the decrease in *D* is a combination of Pol II termination and increasing elongation rate, while the acceleration proposes that the decrease in *D* is primarily a consequence of increasing elongation rate. (**B**) The steady-state GRO-seq density (*D*) and inverse elongation rates ($v^{-1}$) from average transition points of the FP measurements. (**C**) Stages of transcription determining the mRNA level following the productive elongation stage. (**D**) Correlation plot between GRO-seq gene body density as a measure of nascent transcription and RNA-seq as a measure of mRNA steady state level. (**E**) Non-linear correlation between nascent transcription level and mRNA level in highly transcribed genes. To determined the monomial degree of the correlation, a LOESS fit was used for the scatterplot in **D**, and the slopes of the LOESS fit in the higher 50 percentile and the lower 50 percentile were derived. (**F**) Elongation rate correlates with GRO-seq density. Correlation plot was determined from the z-scores of the elongation rates and the gene body GRO-seq densities. (**G**) Correlation of mRNA steady state level and GRO-seq of the 938 mid elongation rate genes (12.5–25 min), and the monomial degree of the correlation derived from the slope. (**H**) Correlation plot of the mRNA production rate (GRO-seq density multiplied by the elongation rate) and mRNA steady state level of the same genes, and the monomial degree of the correlation.

The following figure supplements are available for figure 4:

**Figure supplement 1**. Monte-Carlo modeling of the acceleration and the termination hypotheses.

downstream of TSS). This indicates that the bulk of the change in Pol II density stems from the acceleration of stably elongating Pol II molecules.

We further assessed the contribution of termination and acceleration by simulating these three models of Pol II dynamics and density profiles with different levels of termination. The measured

elongation rates fit better to the model when the termination was insignificant compared to the acceleration (*Figure 4—figure supplement 1*). Therefore, Pol II termination does not appear to have a global contribution to the amount of Pol II in the regions more downstream of 2–3 kb from the TSS.

## Pol II elongation rate partially explains steady-state mRNA level

Elongation rates can influence the total mRNA output. The steady state mRNA level is determined by the balance of nascent RNA production rate, which is represented by GRO-seq density times the elongation rate, as well as RNA processing and mRNA degradation (*Figure 4C*). Thus, deviation from a log–log slope of 1 in a correlation plot between steady state mRNA levels as measured by RNA-seq and GRO-seq density (*Figure 4D*) could potentially be explained by differences in elongation rates. Interestingly, genes that are relatively highly expressed show a slope of 1.8 indicating that mRNA level is not a simple function of GRO-seq density for these genes (*Figure 4E*). We observed a modest correlation between elongation rate and GRO-seq (*Figure 4F*), indicating that highly active genes have faster elongation rates and produce more mRNA, in agreement with a previous analysis of a smaller set of inducible genes (*Danko et al., 2013*).

To further assess the influence of elongation rates on steady state mRNA level, we examined the correlation between mRNA-seq and GRO-seq in the large group of mid elongation rate genes (n = 938). These highly expressed genes display a log–log slope of 2.2 (*Figure 4G*). In contrast, when plotting the production rates, that is the mid elongation rates multiplied by the corresponding GRO-seq densities, vs mRNA level, the slope is reduced to 1.6 (*Figure 4H*), indicating that elongation rate can partially explain the divergence between Pol II density and the mRNA level at highly transcribed genes. The remaining positive correlation might be explained either by increased mRNA stability or RNA processing efficiency in highly transcribed genes.

## Determinants of the elongation rates

The large number of measured elongation rates allows us to identify major factors that influence variation in elongation rates. We used genes for which we measured the large number of mid elongation rates (12.5–25 min, n = 938) to increase statistical power, and to rule out major effects of intrinsic acceleration in the comparison between elongation rates. Existing genome-wide data of transcription factors, chromatin modifiers, and chromatin modifications were used to detect determinants of elongation rate (see for full list of features *Figure 5—source data 1*).

### Chromatin marks

To understand how chromatin could modulate elongation rate, we first searched for correlations between various existing ChIP-seq data and elongation rates, both in the promoter and in the region where elongation rates were measured. For factors showing more significant correlations, we then made density profiles of ChIP-seq data in genes ordered by increasing mid-elongation rate (*Figure 5A*, left). Interestingly, H3K36me3 and CpG methylation were depleted over larger regions around the TSS in faster genes, while H3K4me1 and H3K79me2 were enriched in these same regions (*Figure 5A*, right). CpG methylation and H3K36me3 are enriched at exons, while H3K79me2 levels decrease upon encountering the second exon (*Schwartz et al., 2009*; *Huff et al., 2010*; *Maunakea et al., 2013*). Therefore, we examined the correlation between elongation rate and the length of intron 1 (*Figure 5A*). Indeed, the elongation complex tends to travel faster in genes with larger first introns.

### Exon density

The positive correlation between intron 1 size and elongation rate shows that the absence of exons has a positive effect on elongation. Therefore, we plotted the mean elongation rates of genes based on the number of exons within the average region that Pol II travels between 12.5 and 25 min FP (7.5–30 kb) (*Figure 5B*, left). Interestingly, the striking negative linear correlation between the number of exons and the elongation rates suggests that Pol II slows whenever it encounters an exon. To quantify the delay per exon, we plotted the exon density within the transition region of each gene against the elongation rate, and added a direct linear regression (*Figure 5B*, right, *Figure 5—figure supplement 1A,B*). Addition of an exon within the three regions, in which early, mid, and late elongation rates were derived, results in average delays of 37, 31 or 23 s respectively. Thus, the co-transcriptional delay of Pol II at exons is striking and might be linked to coupled splicing events. Interestingly, the delay is somewhat less in downstream regions of the gene

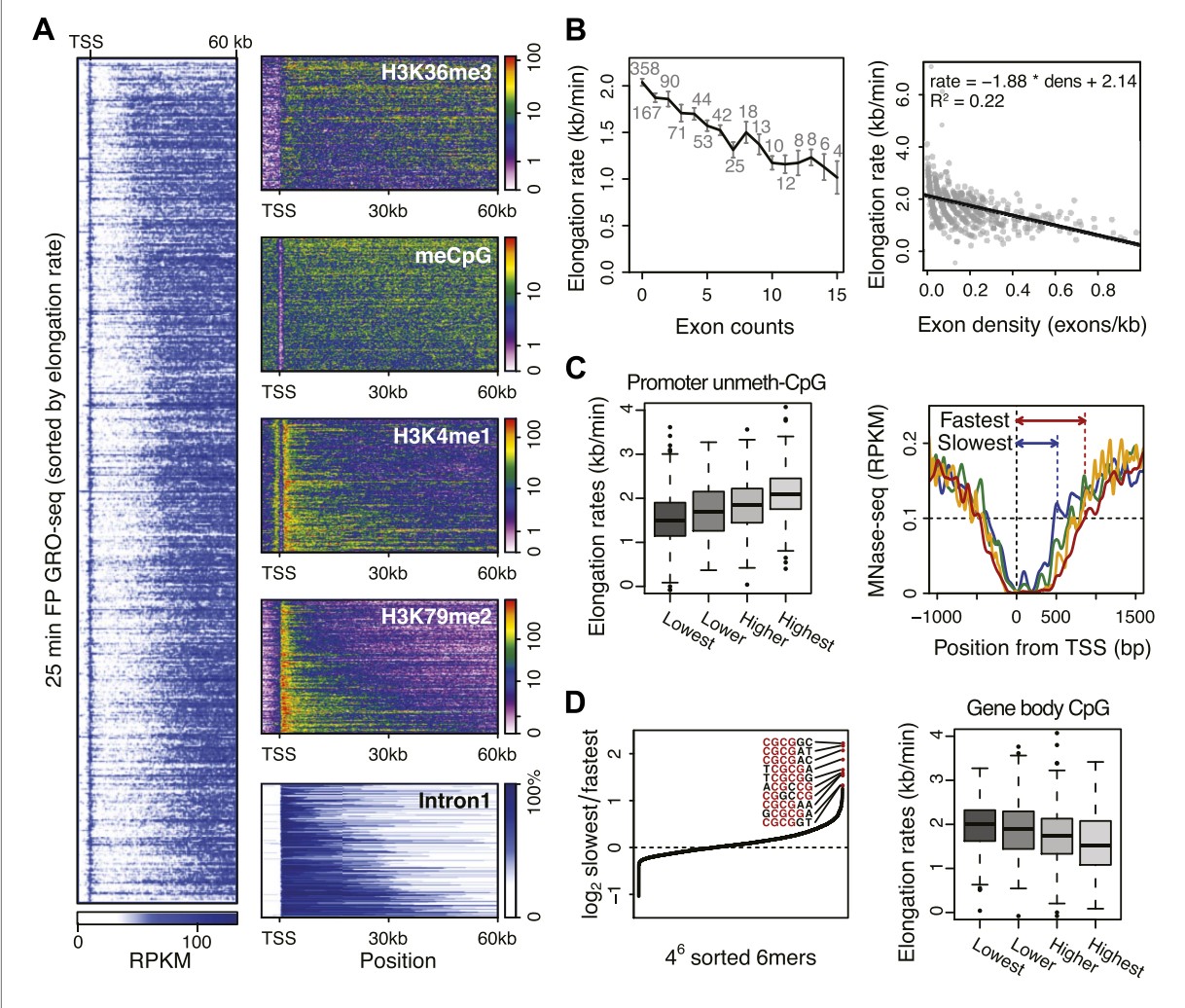

**Figure 5**. Genome-wide features that associate with the elongation rates. (**A**) Profile of the 25 min FP GRO-seq data in mid elongation rate genes (n = 938), ordered by increasing rate (left panel). Right panels depict genomic features in the same gene order (but more compact) from top to bottom: H3K36me3, CpG methylation, H3K4me1, H3K79me2 and intron1 size. (**B**) Elongation rates strongly correlate with exon density. (Left) Mean elongation rates of mid rate genes plotted as a function of the exon count within the average mid transition region (12.5–25 min, or 7.5–30 kb). (Right) Elongation rate as a function of exon density in individual mid transition regions of genes with 1 or more exons, with linear regression. (**C**) (Left) Boxplots of the elongation rates in quartiles of unmethylated CpG content at promoters. (Right) Nucleosome pattern around the TSS for rate quartiles; Nucleosome density at promoters correlates with downstream mid elongation rates. (**D**) CpG content reduces elongation rate. (Left) 6-mer combinations from 'AAAAAA' to 'UUUUUU' ordered by relative occurrence in the slowest quartile of the mid-elongation rate genes compared to the fastest quartile (n = 234 each). (Right) Boxplots of the elongation rates, in quartiles of CpG content in the average mid transition regions of genes.

The following source data and figure supplements are available for figure 5:

**Source data 1**. ChIP-seq datasets used in this study.

**Figure supplement 1**. Effects of exon density and CpG on elongation rates.

body when Pol II is more fully phosphorylated (*Figure 3—figure supplement 3F*) and is generally elongating faster (*Figure 3*).

## Promoter structure
Somewhat surprisingly, promoter architecture seems to have an effect on downstream elongation rates. We found a striking positive correlation between CG content at promoters, that is CpG

islands, and the downstream mid elongation rates (*Figure 5C*, left). Unmethylated CpG islands are often associated with broad transcription initiation (*Core et al., 2008*), likely because CpG islands deter nucleosome binding (*Ramirez-Carrozzi et al., 2009*; *Fenouil et al., 2012*). Interestingly, nucleosomes can be barriers to Pol II elongation and cause transient pauses to delay the Pol II traversal (*Hodges et al., 2009*; *Bintu et al., 2012*). Therefore we explored the nucleosome occupancy as a function of elongation rates, and found that the nucleosome free region (NFR) is wider in genes with faster elongation rates (*Figure 5C*, right); the fastest gene group is shifted approximately 2 nucleosomes downstream compared to the slowest genes. This shows that a more nucleosome-free promoter architecture correlates positively with elongation rates, and implicates an intrinsic component of Pol II rate is determined in the earliest stages of transcription.

## DNA sequence

To assess if CG content within the gene body has an effect on elongation, we examined the DNA sequences in the average mid elongation rate 7.5–30 kb region by searching for differences in all possible 6-mer DNA sequences between the fastest and the slowest groups of genes. Interestingly, the top 20 6-mer sequences associated with slower elongation rates were enriched for repeats of CG dinucleotides (*Figure 5D*, left). Moreover, we found a modest negative correlation between the CG density and the elongation rates in individual genes (*Figure 5—figure supplement 1C*), opposite of the effect CG content has at the promoter. Interestingly, while elongation rates seems to be positively related to the NFR and GC content at promoters, elongation rate correlates negatively with CG content within the gene body (*Figure 5D*, right panel) and is unaffected by nucleosome occupancy (*Figure 5—figure supplement 1D*). This negative effect also seems independent of exonic enrichment for CG content and nucleosomes (*Schwartz et al., 2009*; *Tilgner et al., 2009*), as we still observed the correlation between elongation rates and CG content in intronic regions (*Figure 5—figure supplement 1C*). The increased melting temperature of GC-rich DNA may explain this general reduction of elongation rates (*Nechaev et al., 2010*), but the methylation status of the CG dinucleotides (*Figure 5A*) may also lead to differential regulation of Pol II elongation rates.

## General model for the elongation rates

As described above, gene-to-gene differences of the elongation rates are associated with various features of genes. Many of these are tightly linked to each other; the first intron length is related to H3K79me2 mark and directly linked to the exon density; CpG methylation and H3K36me3 is enriched specifically in exons. To ascertain the primary from secondary determinants of the elongation rates, we used a multivariate regression analysis. For individual genes, each feature was scaled to a rank-order based z-score to assume a Gaussian distribution, making different metric types comparable. We then iteratively assessed how combinations of the features fit to the linear model, and determined the association network with our elongation rate measurements.

Out of many features associated with the gene body and promoters of mid elongation rate genes (*Figure 6A*), we examined further those features that correlated most with elongation rates (*Figure 6A*). All of these features are significantly associated with the elongation rates, the exon density having the strongest correlation (*Figure 6A*). To find which of the features can have an independent effect on elongation rates, we examined how much of the correlation between each feature and the elongation rates could be explained by another feature (*Figure 6B*). For example, the correlation between elongation rates and intron 1 length, gene body H3K36me3, and H3K4me1 can be explained almost entirely by the exon density alone (*Figure 6B*), since they do not show correlation with the residual of the elongation rates after fitting to the exon density model (*Figure 6—figure supplement 1A*). However, gene body H3K79me2 and gene body CpG features cannot be explained mostly by exon density (*Figure 6—figure supplement 1A*). Likewise, H3K79me2 and CpG content have independent components from each other, while CpG methylation is tightly associated with CpG content (*Figure 6B*). These tight associations between exon density and intron 1 lengths, and between CpG count and methylation are also shown by principal component analysis (*Figure 6—figure supplement 1B*), which also reveals the contribution of less major factors such as cohesin and the promoter architecture characterized by unmethylated CpG islands and the NFR. Therefore, the multivariate model progressively explains more of the variances and shows better fits when CpG content, exon density, and H3K79me2 are combined to explain elongation rate (*Figure 6C*). Addition of more factors only

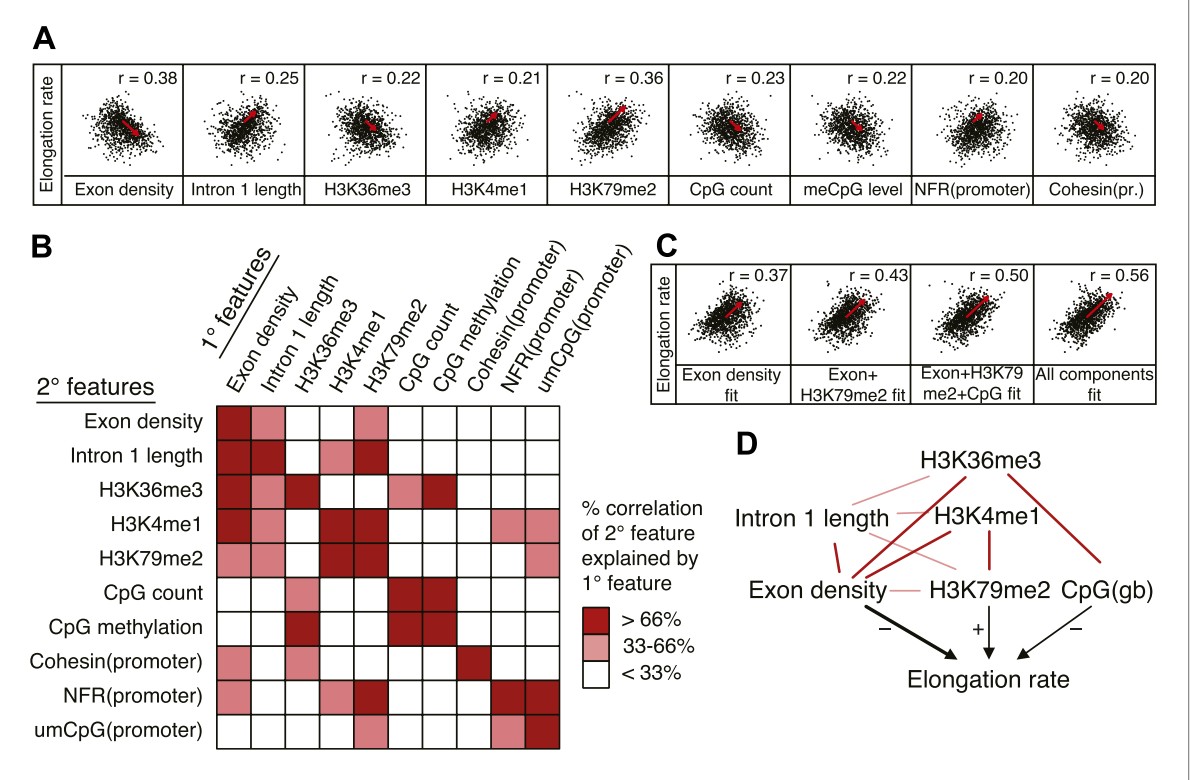

**Figure 6**. Variations in elongation rates are largely explained by exon density, H3K79me2 and CpG content. (**A**) Most prominent correlations between mid elongation rates (n = 938) and various features of mid rate genes after transformation to a rank-order based z-score assuming a Gaussian distribution. (**B**) Iterative assessment to establish how much of each feature in **A** is explained by a second feature from **A**. If a 1° feature explained >66% of the correlation of a 2° feature, the 2° feature was deemed dependent of the 1° feature, and did not contribute individually to the elongation rate model. (**C**) Correlation of the elongation rate with the linear prediction model using exon density only (first panel), exon density/ H3K79me2 (second panel), exon density/H3K79me2/CpG content (third panel), and all components (last panel) shows that the exon density, H3K79me2 and CpG content explains most variation in elongation rates, and that addition of other components improves the correlation only slightly. (**D**) Interaction plot of major gene body features that influence elongation rates, either directly, or indirectly. Color key of (**B**) is applicable to the color of the interaction lines, the boldness of the arrows indicates individual contribution to elongation rate modulation, and the plusses and minuses indicate a positive or negative effect.

The following figure supplements are available for figure 6:

**Figure supplement 1**. Comprehensive linear models of the elongation rates using multiple genomic features.

slightly increases the variance explained by the model of elongation rates (*Figure 6C*, right panel), indicating that CpG content, exon density and H3K79me2 are the main modulators of elongation rate. While CpG content is likely to exert effects throughout the gene, H3K79me2 is mostly enriched within exon and intron 1, indicating that this histone modification is beneficial to elongation early in the gene, while the elongation complex is still accelerating. Together, multiple interactions of chromatin and gene features together impact elongation rate, and provide a first comprehensive examination of the regulation of elongation rate (*Figure 6D*).

## Discussion

In this study we emphasize the role of elongation rate in the modulation of transcription in mESCs. While transcription elongation rates often seem to be taken for granted, we show that elongation rates are surprisingly diverse, both at the promoter proximal region, within the gene body, and between genes. We definitively show that all genes have the potential of being regulated at the promoter proximal pause site, as all initiated Pol II undergoes a P-TEFb dependent step of escape into productive elongation. Furthermore, we find that the average half-life of paused Pol II is 7 min,

indicating that Pol II pausing is remarkably stable, and not the consequence of fast, repeated rounds of initiation and termination. Finally, we measured elongation rates ranging from 0.5 to 4 kb/min at ~1000 genes and within different regions of genes. Strikingly, Pol II accelerates as it transcribes through the gene body, and a large amount of variation in elongation rates occurs between genes, and is associated with chromatin and exon density. Overall, we are able to predict ~30% of the intergenic variation in elongation rates with just three features, exon density, CpG content and methylation, and H3K79me2, providing a first quantitative assessment of how elongation rates are affected by gene structure.

## Promoter proximal pausing and P-TEFb dependent escape are integral parts of the transcription cycle

Inhibition of P-TEFb, like inhibition of initiation, affects >95% of actively transcribed genes, showing that escape from pause is an integral step in the transcription cycle. However, not all active genes show signs of promoter proximal accumulation of paused Pol II in mES cells (*Min et al., 2011*, this study). Clearly, this accumulation of Pol II depends on both the rate with which pause complex is formed (the transcription initiation rate) and its rate of escape to elongation. Some transcription factors stimulate the formation of paused Pol II, others recruit P-TEFb directly or indirectly to facilitate the escape of paused Pol II, while others do both (*Blau et al., 1996*; *Rahl et al., 2010*; *Danko et al., 2013*; *Li and Gilmour, 2013*). The balance of these factors at promoters, together with pausing factors that stabilize Pol II (*Yamaguchi et al., 1999*; *Lee et al., 2008*; *Kwak et al., 2013*) determine the relative accumulation paused Pol II. As we (*Figure 2*) and others (*Henriques et al., 2013*; *Buckley et al., 2014*) have shown, pausing of Pol II is remarkably stable at many genes, indicating that termination does not have a great effect on promoter proximal Pol II levels. This stably-paused Pol II may help maintain accessibility of the promoter to regulatory factors that tune transcriptional output by simply changing the escape rate of paused Pol II into the body of the gene. Indeed, heat shock has recently been shown both optically and biochemically to increase the escape of paused Pol II into the gene body at the *Drosophila Hsp70* locus without decreasing termination of paused Pol II, arguing against a role for termination in the regulation of the accumulation and escape of paused Pol II.

If Pol II undergoes a pausing step in the transcription cycle at all genes, then blocking P-TEFb kinase activity should lead to an increase of paused Pol II. Indeed, this is what we observe for the majority of genes (*Figure 2*). However, a small fraction of poorly expressed and weakly paused genes show a reduction in paused Pol II after FP treatment. We speculate that transcription initiation, Pol II pausing, and escape into productive elongation are not necessarily uncoupled processes for all genes. Instead, disruption of pausing dynamics by FP may feed back on transcription initiation and decrease the formation of paused Pol II. Other observations support a cross-talk of pausing and initiation. Depletion of Nelf can decrease expression of a subset of genes, indicative of a dependency of initiation on pausing, in this case through the maintenance of nucleosome free promoters by paused Pol II (*Gilchrist et al., 2008*). Furthermore, precise mapping of Pol II reveals that core promoter elements appear to contribute to stronger and more promoter proximal pausing in *Drosophila* (*Kwak et al., 2013*; *Venters and Pugh, 2013*). Overall, the specific inhibitors used in this manuscript inhibit either initiation or pause escape seemingly with few secondary effects, as the Pol II inhibition profile within gene bodies is remarkably similar after either drug treatment. Nonetheless, perturbations of one process could possibly have effects on the other, at least for some genes.

## Causes and consequences of elongation rate variation between cells

Because P-TEFb inhibition affects all active genes, we could determine over 1300 individual elongation rates of Pol II, often at multiple regions within the same gene. *Danko et al. (2013)* has used GRO-seq following rapid induction of transcription to examine the rate of the Pol II induction wave front. Our elongation rates derived after inhibition of P-TEFb were slightly lower in comparison to the elongation rates derived after induction which may point to a model in which lagging Pol II pushes leading Pol II thereby increasing overall elongation rates (*Saeki and Svejstrup, 2009*). Indeed, elongation rates in highly expressed genes, where elongating Pol II is more tightly packed, are faster compared to rates in poorly expressed genes in both studies (*Figure 4*; *Danko et al., 2013*).

Moreover, we have shown that chromatin composition, specifically H3K79me2 and CpG content, of genes influences elongation rates. Interestingly, mouse ESCs have high levels of DNA methylation

(*Stadler et al., 2011*). DNA methylation and CG content negatively influence elongation rate, and could partially explain the difference between mESC and differentiated cell line derived elongation rates (*Danko et al., 2013*). Finally, repression of H3K79me2 has recently been observed to increase reprogramming efficiency of fibroblasts to iPSCs, by reducing expression of lineage specific genes in the first stages of reprogramming (*Onder et al., 2012*). We have shown that elongation rates can influence steady state mRNA levels (*Figure 4*), and that H3K79me2 positively influences elongation rate (*Figure 6*). We therefore argue that reprogramming efficiency by reducing H3K79me2 levels could be a result of elongation rate modulation.

Finally, we emphasize that the variation of elongation rates revealed in this study have enormous potential in timing of mRNA production. Mammalian genes of over 200 kb and twofold differences in elongation rates between genes are not uncommon, leading to variation in response time of mRNA production of an hour or more. This timing can be very relevant in development and the stress response (*Thummel, 1992*; *Swinburne and Silver, 2008*), and could not only be dependent on intron length, but also be regulated by elongation rate modulation.

### Elongation rates and splicing

The hypothesis that Pol II elongation rate regulates splicing is longstanding (reviewed in *Shukla and Oberdoerffer 2012*). Here, we show directly that exon density is the greatest predictor of elongation rates (*Figure 6*), strongly suggesting that Pol II slows down at each exon. Exonic features such as CpG methylation, H3K36me3 and H3K4me1 could work synergistically to establish a transient slow down at exons (*Figure 6*). Although we demonstrate that Pol II slows down at exons, presumably to facilitate splicing, we could not determine whether a slowly transcribing Pol II increases inclusion of exons. Although rates of elongation have been shown to influence splicing outcomes (*Howe et al., 2003*; *de la Mata et al., 2003*), the effects on specific genes, we suggest, will be governed by competing processes such as rates of splicing complex assembly, RNA secondary and tertiary structure formation, and regulatory factor binding. While our data cannot on its own assess the outcomes on alternative splicing of such competing and cooperating processes, our observed slow down of transcription at exons supports the general view that Pol II elongation rates are coupled to splicing at all exons.

## Material and methods

### Cell culture, nuclei isolation and GRO-seq library preparation

Cell culturing of the V6.5 mES cell line was done as in *Monkhorst et al. (2008)*, and drug treatment was performed on pre-plated mES cells to remove irradiated MEF-feeder cells, grown for one passage on 15 cm$^2$ plates up to ~70% confluence before isolation of nuclei. Drugs treatment was done by replacing ES medium with pre-heated ES medium containing 300 nM FP, 500 nM Trp or 0.0125% DMSO as no Trp control. Nuclei isolation was done according to *Min et al. (2011)*. Nuclear run-on and nascent RNA library preparation was performed as in *Core et al. (2008)*. In brief, after rinsing the 15 cm$^2$ plates with drug-treated cells with ice-cold PBS, pH 7.4, cells were scraped off in 15 ml cell lysis buffer (10 mM Tris-Cl, pH 7.5, 300 mM Sucrose, 3 mM CaCl$_2$, 2 mM MgAc$_2$, 0.5% NP-40, 5 mM DTT, 1 mM PMSF, protease inhibitors), and spun down at 4°C. Cells were dounced 50x in 5 ml fresh cell lysis buffer on ice and spun down, after which supernatant was discarded and nuclei were taken up in ~250 µl of glycerol storage buffer (50 mM Tris-Cl, pH 8.3, 40% glycerol, 0.1 mM EDTA, 5 mM MgAc$_2$, 5 mM DTT, 1 mM PMSF, protease inhibitors) and snap frozen.

For each nuclear run-on (NRO), $10^7$ nuclei were mixed with an equal volume of reaction buffer (10 mM Tris-Cl pH 8.0, 5 mM MgCl2, 1 mM DTT, 300 mM KCL, 20 units of SUPERaseIn, 1% sarkosyl, 500uM ATP, GTP, and Br-UTP, 2 µM CTP and 0.33 µM α-32P-CTP [3000 Ci/mmole]). The NRO was performed at 30°C for 5 min, and a population of ~100 different in vitro transcribed *Arabidopsis thaliana* spike-in RNAs with and without Br-UTP was added to the nascent RNA. The RNA was fragmented to ~150 nts with 0.2N NaOH and BrU-RNA was isolated three consecutive times with BrdU-antibody beads (sc-32323; Santa Cruz Technologies, Dallas, TX), with enzymatic TAP and PNK treatments to remove the cap and 3'-phosphate and to add a 5'-phosphate, as well as Illumina adaptor ligations between the BrU-RNA isolation steps. The three consecutive isolation steps lead to an approximate 500.000x enrichment of BrU-RNA over background RNA. BrU-RNA was reverse transcribed, amplified, barcoded and Illumina sequenced. Each dataset was done in replicate.

## Sequence alignment, normalization and analysis of FP and Trp drug treatment effects

All the GRO-seq libraries were sequenced in 50 nt runs on the Illumina HiSeq and split by barcode. Reads were trimmed to 32-mers and Illumina adaptors were removed with the cutadapt tool (https://code.google.com/p/cutadapt/) and aligned uniquely with two mismatches with bowtie to the mm9 reference genome. Replicates were highly correlated and were pooled for further analysis (*Figure 1—figure supplement 1*), with exception of the extensively degraded 25 min Trp-treated #1 replicate. Normalization between datasets was done with uniquely aligned spike-in RNA reads. Sequence datasets can be found under GEO admission number GSE48895.

The mm9 RefSeq genelist was used as reference genelist for all analysis. Unmappable regions of the genome were identified and excluded by aligning the genome to itself in 30-mers and reads aligned to these regions were not used in analysis. To establish which genes were active above background with a Fisher exact p-value of <0.05, we mapped reads from control datasets to gene-poor regions, took 10 × the average read density of these two datasets as a safe threshold for background (5 × 10$^{-4}$ reads/bp). The genes for analysis of FP and Trp sensitivity are the top 75% active genes larger than 3.5 kb present in both the FP as the Trp control dataset, without genes that have an annotated TSS within 1000 bp on the opposite strand (bidirectional genes) and genes that have an annotated polyA site upstream of its TSS within 10 kb (tandem genes) (*Figure 2—figure supplement 1A*, n = 6380). Pause and divergent peak locations were found by searching for maximum sense or antisense strand read density in 10 bp windows from ±500 bp or −1000 to +500 around the annotated TSS of the 6380 selected genes, respectively. The peak was defined as a 250 bp region centered on the maximum 10 bp window. The pausing index is the ratio of the pause peak density and the annotated gene body region density (+1 kb from the TSS to polyA site). To calculate the fold change after FP or Trp treatment for each timepoint, we added a pseudo-count to the divergent, pause or gene body region close to the TSS (+1 kb to +3.5 kb only), calculated the density in the mappable region of each and took the ratio of the read densities. The change was significant if the Fisher Exact p-value was <0.05.

Decay rates were calculated by selecting seven points randomly of the seven Trp datasets for 1000 times and doing non-linear regression using an exponential decay equation ($R_t = R_{WT} × e^{-(\lambda t)}$, with pause peak read density R, decay rate λ, and Trp treatment time t) with each of the seven points. The mean of the regressions is the decay rate and the standard deviation of the decay rate is the standard deviation of the 1000 individual regressions.

Composite profiles of all genes >12.5 kb or >150 kb were made in R with read density taken in 50 bp windows. Density plots of the 6380 selected genes around the TSS were made by taking the log$_{10}$ of counted reads plus a pseudo-count in 10 bp windows around the annotated TSS for the sense and antisense strands in the control and 50 min treated datasets. The change in each 10 bp window was calculated by subtracting the no FP/Trp from the 50 min FP/Trp log$_{10}$ read count. Genes were ordered by maximum pause peak read density decrease or increase after 50 min Trp or FP treatment. The density in windows was plotting using the R packages gplots and RColorBrewer.

## Estimation of elongation rates using hidden Markov model (HMM)

First, we selected genes longer than a sufficient cut-off for each time-point (30 kb for 5 min and 12.5 min, 60 kb for 25 min, and 150 kb for 50 min), that have corresponding transcription units at the annotated TSS, but do not contain intragenic transcription units defined by the genome-wide transcription unit calling algorithm (described below in this section). Also, genes that have premature termination before the annotated 3′ ends and/or the 60 kb/150 kb mark are removed using a regional transcription unit calling algorithm (described below in this section). After filtering, the number of selected genes are n = 4461, 2769 and 571 for genes longer than 30 kb, 60 kb, and 150 kb, respectively. The transition points from the drug affected (inhibited) region to the drug unaffected (uninhibited) region of the gene body were determined by a regional transition point calling algorithm (described below in this section) for each replicate of FP or Trp GRO-seq time courses. The following is the description of the custom made HMM algorithms in c++.

### Regional transition point calling algorithm

For the selected genes, GRO-seq read counts from the TSS to 60 kb or 150 kb positions are binned for each time-point, and were divided by the untreated read counts at the same bins (*Figure 3—figure*

*supplement 1A*). The bins sizes are 500 bp, 1 kb, 2 kb, and 5 kb respectively for analyzing 5 min, 12.5 min, 25 min and 50 min time points. These sizes were selected so that each gene will have approximately 30–40 bins throughout the gene body regions, which was the optimal number for the HMM results. For each gene, the ratios in individual bins were internally normalized, by dividing by the average ratios of the last five bins at the 3′ ends. These bins are considered as the Markov process as the observations (*Figure 3—figure supplement 1A*). For efficient HMM calculation, the normalized ratios were digitized, ranging from 0 to 2.0, with a step size of 0.05. Therefore, each binned position can have 40 observed states of the read ratio, and the probability of each state follows a binominal distribution of $B(n, e)$, where $n = 20$ and $e$ is the emission probability. We assumed two hidden states, 'inhibition affected' and 'inhibition unaffected', with two emission probabilities $e_1$ and $e_2$. The transition probabilities between the 'affected' and 'unaffected' states are $p_{11}$, $p_{12}$, and $p_{21}$, and are unidirectional. The Baum-Welch algorithm is used to estimate the transition and emission probabilities, by iterative calculations until the probabilities converge. If the iteration is over 200 cycles without convergence, we dropped the gene from further analysis. The transition point is calculated from these probabilities.

## Genome-wide transcription unit calling algorithm

The untreated GRO-seq data is used similarly as the regional transition point calling algorithm with some modifications. First, instead of using ratios relative to the reference dataset, we used binary observation values for each 200 bp bin of the Markov process; 1 if there is a GRO-seq read within the bin and 0 if there are no reads. In this case, the binomial distribution becomes $B(2, e)$. Second, the full length of each chromosome was used rather than individual genes. Third, the Baum-Welch algorithm was allowed to run up to 1000 iterations. Finally, a Viterbi path was calculated to define transcriptionally active regions.

Using this de novo transcription unit (TU) calling, we selected TUs that have both sense and divergent pairs starting within 2 kb from each other. These paired TUs can indicate annotated TSSs, unannotated TSSs, or other regulatory transcription activities such as lncRNA or enhancer transcripts. We compared these TUs to the annotated long genes and removed the genes that did not have paired TUs near the annotated TSS (<2 kb). Also, we dropped genes that contain divergent TUs within the gene body, since these paired sense TUs can indicate alternative start sites that may interfere with detection of the inhibition wave by the regional transition point calling algorithm.

## Regional transcription unit calling algorithm

Using untreated GRO-seq data, we generated binary bins from TSSs defined by the genome-wide transcription unit calling algorithm to 50 kb downstream of the annotated 3′ end in each individual gene, and used these values to estimate the probability parameters of the HMM. This is similar to the genome-wide transcription unit calling algorithm in that it uses binary observation values, but also similar to the regional transition point calling in that it is done in individual genes. One difference is that it starts from the active region and detects the transition point into the inactive region or in other words the end of the TU, which is opposite of the regional transition point calling algorithm. Genes that have the transition point before the 60 kb/150 kb point are dropped, as this may interfere with detection of the inhibition wave by the regional transition point calling.

## Relative elongation rate estimate using nascent RNA-seq

We analyzed relative elongation rates with methods described in *Ameur et al. (2011)*, using intronic reads of total poly-A(−) RNA-seq (*Sigova et al. 2013*). Intronic reads are sparse, so we pooled multiple genes to assess the relative elongation rate. First, introns longer than 10 kb are selected. Introns containing annotated alternative exons were split in a 5′ and 3′ intron to exclude the interference from exonic RNA-seq reads. Second, selected introns are subgrouped by length in 10 kb bins; *that is* 10–20 kb, 20–30 kb, 30–40 kb groups, *etc*. We used up to 50 kb for the estimation of elongation rates (up to 80 kb are plotted). Third, introns are aligned at the 3′ splicing sites (3′SS), and intronic reads are counted in 100 bp windows in each intron length subgroup. To normalize for differences in the transcription level in each group, the read counts in windows are normalized to the read density 1 kb upstream of the 3′ SS to the 3′SS in each group (*Figure 3—figure supplement 2B*). Finally, we plotted read counts per window for each intronic position in all the subgroups (*Figure 3D*). The slope of the RNA-seq gradient was obtained using the linear regression in the R statistics package.

## Kinetic Monte Carlo simulation of the acceleration and the termination models

Apart from the termination and acceleration models described in *Figure 4A*, we also considered a mixed model, in which elongating Pol II could consist of different populations; fast Pol II, transcribing at maximum speed throughout the entire gene body, and slow, non-processive Pol II. If the slow population terminates prematurely leaving the fast Pol IIs, then the overall apparent elongation rate would increase as transcription proceeds in the gene body. To differentiate between the termination, acceleration and mixed model, we used a Monte Carlo simulation describing the dynamics of Pol II movement through the gene body. In short, we simulated the time course of the inhibition wave of Pol II (*Figure 4—figure supplement 1A*) and used the regional transition point calling algorithm (*Figure 3—figure supplement 1A*) to define the transition points of inhibition waves and elongation rates. The relationship between the simulated Pol II density and the inverse of average elongation rates of 1000 simulated experiments shows a clear difference between the models (*Figure 4—figure supplement 1B,C,D*). When we plotted the slope of simulated elongation rates vs the slope of inverse density in various simulated acceleration and termination models along with actual observations, the observations appear to fit better with the distribution of the acceleration models than the termination models (*Figure 4—figure supplement 1E*).

In detail, the dynamics of elongating Pol II in gene body is simulated using a newly designed modeling program to describe Pol II transcription through a gene. First, we modeled a Pol II transcription complex entering the gene body region with an entry rate ($r$) as a function of time. For the steady state assumption, $r$ is a constant over time ($t$), while for the simulation of the inhibition wave $r$ is an exponential decay function of $t$.

Each Pol II molecule was generated with the rate $r$, and has a randomly assigned activity parameter ($A$), from 0 to 100 as a percentile. This activity parameter $A$ is an intrinsic value that determines the relative elongation and the termination rates for each Pol II molecule. The termination constant ($k_t$) and the elongation rate ($v$) are the functions dependent on the activity ($A$) as well as the position ($x$) within the gene body of the Pol II molecule. For instance, in a simple acceleration model where all Pol II molecules accelerate uniformly, $k_t = 0$ for all $A$ and $x$, while $v$ is an increasing function of $x$ but a constant function for $A$. In a termination model, intrinsically active Pol II molecules elongate faster while less active ones elongate at a slower rate and terminate more frequently. In this case, $k_t$ is a decreasing function of $A$, and $v$ is an increasing function of $A$ regardless of $x$.

When running a simulation, the entry, termination, and progression events are assessed after each time increment of $\Delta t$. For each event, a pseudorandom number between 0 and 1 is generated and compared to the probability of initiation as described by $1-exp(-r\Delta t)$, termination as described by $1-exp(-k_t\Delta t)$, and processive elongation as described by $1-exp(-v\Delta t)$ respectively. If the number is less than the probability of any of the processes, the status of the polymerase changes accordingly. For the approximation of the progression event, the polymerase can move $k$ bases following the Poisson distribution if the pseudorandom number is in the range ($F(k; \lambda)$, $F(k+1; \lambda)$), where $F$ is the cumulative Poisson distribution function and $\lambda = v\Delta t$. If there is a collision event between two polymerases, the leading polymerase terminates.

The distribution of simulated Pol II in N = 1000 DNA templates are equilibrated for 10,000s. The average Pol II distribution at this point is recorded as $D$. Upon the simulation of the decay of entry, average Pol II distribution is recorded every 100 s over 100,000 bp region. The average distribution at each time point is analyzed using the HMM, and the transition points are estimated. From the time course of transition points, the apparent elongation rates ($v_a$) are calculated as a function of the position ($x$). The slope plot (*Figure 4—figure supplement 1E*) is generated by calculating $\Delta D^{-1}/\Delta x$ and $\Delta v_a/\Delta x$ using linear interpolation between x = 5, 15, 25, 35, 45, 55 kb or t = 5, 12.5, 25, and 50 min.

We tested the following parameter spaces for the simulation. Only one examples of the results for each simple termination, mixed, and stable acceleration models are shown in *Figure 4—figure supplement 1B–D*. However, all the described models were used to generate the scatterplot in *Figure 4—figure supplement 1E*.

## Simple acceleration model

$v(x = 0\ kb) = 5\ bp/s$, $v(x = 60\ kb) = 40\ bp/s$ ; Pol II starts at 5 bp/s (300 bp/min) and accelerates to reach 40 bp/s (2.4 kb/min) at the 60 kb position and downstream. The interpolating $v(x)$ values are generated using cubic Bézier functions. The termination rate $k_t$ is 0.

### Simple termination model

$v(A = 0)$ = 5 bp/s, $v(A = 100)$ = 40 bp/s; the interpolating $v(A)$ values are made with the cubic Bézier functions. $k_t(A = 0)$ = 0.002–0.0005 (/s), $k_t(A = 100)$ = 0; the interpolating $k_t(A)$ values are made with cubic Bézier functions. The $k_t(A = 0)$ values are chosen so that the simulated $D$ curve reflects the observed $D$ curve which gradually decreases in regions $x <20$ kb and becomes nearly a constant where $x >20$ kb. The less active (and slower) population of Pol II is mostly terminated within the first 20 kb region with selected $k_t$ parameters. Finally, combinations of $v$ and $k_t$ functions are used.

### Mixed model

Models with mixed acceleration and termination are also tested. $v(x,A)$ is generated as a combination of the $v(x)$ of the acceleration model and $v(A)$ of the termination model. The $k_t$ functions of the simple termination models are used.

### Regional termination model

The termination rate ($k_t$) can also have an added dimension and can be treated as a function of both $x$ and $A$. The interpretation of this is that termination takes place at some preferred positions along the gene. We adjusted the shape of the function so that slower Pol II is relatively stable near the 5′ side of the gene to make the apparent elongation rate slower at the beginning. The increased termination at more downstream position leaves only the faster polymerase populations and the apparent elongation rates are higher.

## Acceleration constants of intrinsically accelerating Pol II

We calculated the intrinsic acceleration constant $a$ of Pol II with $v_{end} = v_{start} \times a \, \Delta t$ for corrected elongation rates between 5 to 12.5 min and 12.5 to 25 min FP treatment, and for 12.5 to 25 min and 25 to 50 min FP treatment. Elongation rates were corrected for exon density within the transition region where the rate was measured by using the linear regression formulas in *Figure 5B* and *Figure 5—figure supplement 1A,B*. For the linear regressions the elongation rates over transition regions that had an exon density >0 were taken into account to assess the additive time delay per exon.

## ChIP-seq analysis and linear modeling of the genomic features

ChIPseq datasets listed in *Figure 5—source data 1* were downloaded and aligned to the mm9 genome when necessary. Read density was derived in the mappable promoter region or the average 12.5–25 min or 25—50 min transition regions. We correlated read density of ChIPseq factors in the promoter or transition regions with elongation rates. For factors that correlated with elongation rates, we made additional density profiles from −2 to 30 kb from the TSS in 25 bp windows in the 12.5–25 min mid elongation rate genes, ordered by mid elongation rate from slow to fast. Also, we looked at the elongation rates in quartiles of correlating factors in the promoter or transition regions.

To apply the linear modeling methods, we converted individual genomic features to fit the Gaussian distributions using rank ordered z-statistics (R statistics package). Briefly, the percentile rank of a gene's specific genomic feature is converted to a z-score through the inverse cumulative Gaussian distribution function. This conversion enables the comparison of different non-linear features, yet also allows us to use the linear modeling tools such as least square methods and principal component analysis. Genes with tied rank orders are randomly assigned to have different ranks. The major features taken into account have a p-value<0.01 from the simple linear regression (function lm() in R). The z-score distribution between the elongation rate and a major feature is plotted in a [−4,4] × [−4,4] range (*Figure 6A*). The principal component analysis (function princomp() in R) between the two features was used to identify the eigenvector that shows the direction of correlation, and a red arrow was depicted to represent the direction as well as the magnitude of the correlation coefficient.

For the iterative linear modeling, we took the residual elongation rates after fitting to the linear modeling using a 1° feature (*Figure 6—figure supplement 1A*, first row), and the residuals are again converted using the rank order z-statistics. The converted residuals are fitted to a 2° feature linear modeling (*Figure 6—figure supplement 1A*, second row). The correlation coefficient of the iterative linear modeling was compared to the correlation coefficient of a single linear modeling using the 2° feature alone, and the percent reduction of the $R^2$ by the 1° feature was color scaled in a combinatorial manner to produce the heatmap in *Figure 6B*. The predicted values from the linear models using one or multiple features are plotted against the actual z-scores of the measured elongation rates to show the convergence of the prediction to the observation (*Figure 6C*).

## Protein fractionation and western blot analysis

Chromatin bound and nucleoplasmic/cytoplasmic free proteins were extracted after treatment of pre-plated mES cells for 50 min with 300 nM FP, 500 nM Trp or 0.0125% DMSO as no Trp control. Cells were rinsed twice in ice-cold PBS, pH 7.4, scraped off the plates, spun and resuspended in nuclei lysis buffer (20 mM Tris-Cl pH 7.5, 3 mM EDTA, 10% glycerol, 150 mM KAc, 1.5 mM MgCl$_2$, 1 mM DTT, 0.1% NP-40, and phosphatase and protease inhibitors), dounced 60 times on ice and centrifuged at 13,000 rpm for 5 min at 4°C. Supernatant was snap frozen as the unbound fraction, while the remaining pellet was resuspended in nuclei lysis buffer and sonicated to break up the chromatin and solubilize the pellet, and snap frozen. Western blot analysis was done in triplicate with antibodies against the Ser5 (3E8; EMD Millipore, Billerica, MA) or Ser2 (3E10; EMD Millipore, Billerica, MA) phosphorylated CTD, or N-terminal Pol II (N-20; Santa Cruz Technologies, Dallas, TX).

## Acknowledgements

We would like to thank Charles Danko, Colin Waters, Leighton Core, Andre Martins, Nicholas Fuda and members of the Lis lab for discussions and critical feedback on the manuscript.

## Additional information

### Funding

| Funder | Grant reference number | Author |
| --- | --- | --- |
| National Institutes of Health | GM25232 | John T Lis |
| Helen Hay Whitney Foundation | | Iris Jonkers |
| Howard Hughes Medical Institute | | Hojoong Kwak |

The funders had no role in study design, data collection and interpretation, or the decision to submit the work for publication.

### Author contributions

IJ, Conception and design, Acquisition of data, Analysis and interpretation of data, Drafting or revising the article; HK, Conception and design, Analysis and interpretation of data, Drafting or revising the article; JTL, Conception and design, Drafting or revising the article

## Additional files

### Major datasets

The following dataset was generated:

| Author(s) | Year | Dataset title | Dataset ID and/or URL | Database, license, and accessibility information |
| --- | --- | --- | --- | --- |
| Jonkers I, Kwak H, Lis JT | 2014 | Genome-wide dynamics of Pol II elongation and its interplay with promoter proximal pausing, chromatin, and exons | GSE48895; http://www.ncbi.nlm.nih.gov/geo/query/acc.cgi?acc=GSE48895 | Publicly available at GEO (http://www.ncbi.nlm.nih.gov/geo/). |

The following previously published datasets were used:

| Author(s) | Year | Dataset title | Dataset ID and/or URL | Database, license, and accessibility information |
| --- | --- | --- | --- | --- |
| Marson A, Levine SS, Cole MF, Frampton GM, et al. | 2008 | Connecting microRNA genes to the core transcriptional regulatory circuitry of embryonic stem cells | GSE11724; http://www.ncbi.nlm.nih.gov/geo/query/acc.cgi?acc=GSE11724 | Publicly available at GEO (http://www.ncbi.nlm.nih.gov/geo/). |

| | | | | |
|---|---|---|---|---|
| Rahl PB, Lin CY, Seila AC, Flynn RA, McCuine S, Burge C, Sharp PA, Young RA | 2010 | Promoter proximal pausing and its regulation by c-Myc in embryonic stem cells | GSE20485; http://www.ncbi.nlm.nih.gov/geo/query/acc.cgi?acc=GSE20485 | Publicly available at GEO (http://www.ncbi.nlm.nih.gov/geo/). |
| Lin C, Garrett AS, Shilatifard A | 2011 | Super Elongation Complex (SEC) and global genomic analyses in murine embryonic stem (ES) cells and in human cells in response to activation signals | GSE30267; http://www.ncbi.nlm.nih.gov/geo/query/acc.cgi?acc=GSE30267 | Publicly available at GEO (http://www.ncbi.nlm.nih.gov/geo/). |
| Chen X, Xu H, Yuan P, Fang F, et al. | 2008 | Mapping of transcription factor binding sites in mouse embryonic stem cells | GSE11431; http://www.ncbi.nlm.nih.gov/geo/query/acc.cgi?acc=GSE11431 | Publicly available at GEO (http://www.ncbi.nlm.nih.gov/geo/). |
| Creyghton MP, Cheng AW, Welstead GG, Kooistra T, et al. | 2010 | Histone H3K27ac separates active from poised enhancers and predicts developmental state | GSE24165; http://www.ncbi.nlm.nih.gov/geo/query/acc.cgi?acc=GSE24165 | Publicly available at GEO (http://www.ncbi.nlm.nih.gov/geo/). |
| Smith ER, Lin C, Garrett AS, Thornton J, et al. | 2011 | The little elongation complex (LEC) regulates small nuclear RNA transcription | GSE32120; http://www.ncbi.nlm.nih.gov/geo/query/acc.cgi?acc=GSE32120 | Publicly available at GEO (http://www.ncbi.nlm.nih.gov/geo/). |
| Ku M, Koche RP, Rheinbay E, Mendenhall EM, et al. | 2008 | Mapping polycomb complexes in human and mouse embryonic stem cells | GSE13084; http://www.ncbi.nlm.nih.gov/geo/query/acc.cgi?acc=GSE13084 | Publicly available at GEO (http://www.ncbi.nlm.nih.gov/geo/). |
| Mikkelsen TS, Ku M, Jaffe DB, Issac B, et al. | 2008 | Genome-wide maps of chromatin state in pluripotent and lineage-committed cells | GSE12241; http://www.ncbi.nlm.nih.gov/geo/query/acc.cgi?acc=GSE12241 | Publicly available at GEO (http://www.ncbi.nlm.nih.gov/geo/). |
| Meissner A, Mikkelsen TS, Gu H, Wernig M, et al. | 2008 | Genome-wide chromatin state maps of ES cells, ES-derived neural progenitor cells and brain tissue | GSE11172; http://www.ncbi.nlm.nih.gov/geo/query/acc.cgi?acc=GSE11172 | Publicly available at GEO (http://www.ncbi.nlm.nih.gov/geo/). |
| Kagey MH, Newman JJ, Bilodeau S, Zhan Y, et al. | 2010 | Control of Embryonic Stem Cell State by Mediator and Cohesin | GSE22562; http://www.ncbi.nlm.nih.gov/geo/query/acc.cgi?acc=GSE22562 | Publicly available at GEO (http://www.ncbi.nlm.nih.gov/geo/). |
| Liu Z, Scannell DR, Eisen MB, Tjian R | 2011 | Control of Embryonic Stem Cell Lineage Commitment by Core Promoter Factor, TAF3 | GSE31270; http://www.ncbi.nlm.nih.gov/geo/query/acc.cgi?acc=GSE31270 | Publicly available at GEO (http://www.ncbi.nlm.nih.gov/geo/). |
| Schnetz MP, Handoko L, Akhtar-Zaidi B, Bartels CF, et al. | 2010 | CHD7 targets active gene enhancer elements to modulate ES cell-specific gene expression | GSE22341; http://www.ncbi.nlm.nih.gov/geo/query/acc.cgi?acc=GSE22341 | Publicly available at GEO (http://www.ncbi.nlm.nih.gov/geo/). |
| Bilodeau S, Kagey MH, Frampton GM, Rahl PB, Young RA | 2009 | SetDB1 Contributes to Repression of Genes Encoding Developmental Regulators and Maintenance of ES Cell State | GSE18371; http://www.ncbi.nlm.nih.gov/geo/query/acc.cgi?acc=GSE18371 | Publicly available at GEO (http://www.ncbi.nlm.nih.gov/geo/). |
| Jiang H, Shukla A, Wang X, Chen WY, et al. | 2011 | Mammalian Dpy-30 regulates genomic H3K4 methylation and is essential for ES cell fate specification | GSE26136; http://www.ncbi.nlm.nih.gov/geo/query/acc.cgi?acc=GSE26136 | Publicly available at GEO (http://www.ncbi.nlm.nih.gov/geo/). |
| Law MJ, Lower KM, Voon HP, Garrick D, Viprakasit V, Mitson M, Zhao Y, Hughes JR, Morris A, Abbott A, Wilder SP, Taylor S, Santos GM, Cross J, Ayyub H, Jones S, Ragoussis J, Rhodes D, Dunham I, Higgs DR, Gibbons RJ | 2010 | A SNF2 protein targets variable copy number repeats and thereby influences allele-specific expression | GSE22162; http://www.ncbi.nlm.nih.gov/geo/query/acc.cgi?acc=GSE22162 | Publicly available at GEO (http://www.ncbi.nlm.nih.gov/geo/). |
| Mendenhall EM, Koche RP, Truong T, Zhou VW, et al. | 2010 | GC-rich Sequence Elements Recruit Polycomb Repressive Complex 2 (PRC2) in ES cells | GSE25197; http://www.ncbi.nlm.nih.gov/geo/query/acc.cgi?acc=GSE25197 | Publicly available at GEO (http://www.ncbi.nlm.nih.gov/geo/). |

| Ma Z, Swigut T, Valouev A, Rada-Iglesias A, et al. | 2010 | Role of Prdm14 in mouse embryonic stem cells: ChIP-seq and RNA-seq analyses | GSE25409; http://www.ncbi. nlm.nih.gov/geo/query/acc. cgi?acc=GSE25409 | Publicly available at GEO (http://www.ncbi. nlm.nih.gov/geo/). |
|---|---|---|---|---|
| Marks H, Habibi E, Brinkman AB, Arand J, Kroeze LI, Kerstens HH, Matarese F, Lepikhov K, Gut M, Brun-Heath I, Hubner NC, Benedetti R, Altucci L, Jansen JH, Walter J, Gut IG, Stunnenberg HG | 2013 | Whole-genome bisulfite sequencing of two distinct interconvertible DNA methylomes of mouse embryonic stem cells | GSE41923; http://www.ncbi. nlm.nih.gov/geo/query/acc. cgi?acc=GSE41923 | Publicly available at GEO (http://www.ncbi. nlm.nih.gov/geo/). |
| Teif VB, Vainshtein Y, Caudron-Herger M, Mallm JP, et al. | 2012 | Genome-wide nucleosome positioning during embryonic stem cell development [MNase-Seq] | GSE40910; http://www.ncbi. nlm.nih.gov/geo/query/acc. cgi?acc=GSE40910 | Publicly available at GEO (http://www.ncbi. nlm.nih.gov/geo/). |
| Sigova AA, Mullen AC, Molinie B, Gupta S, et al. | 2012 | Long non-coding RNAs from divergent transcription of protein-coding genes | GSE36799; http://www.ncbi. nlm.nih.gov/geo/query/acc. cgi?acc=GSE36799 | Publicly available at GEO (http://www.ncbi. nlm.nih.gov/geo/). |

**Reporting standards:** Standard used to collect data: We followed the NCBI GEO database guidelines and repository for submitting and storing the high throughput datasets presented in the manuscript.

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
