## [Decision Letter]

[Editors’ note: although it is not typical of the review process at *eLife*, in this case the editors decided to include the reviews in their entirety for the authors’ consideration as they prepared their revised submission.]

Thank you for sending your work entitled “Genome-wide dynamics of Pol II elongation and its interplay with promoter proximal pausing, chromatin, and exons” for consideration at *eLife*. Your article has been favorably evaluated by a Senior editor and 3 reviewers, one of whom is a member of our Board of Reviewing Editors.

The Reviewing editor and the other reviewers discussed their comments before reaching this decision, and the Reviewing editor has assembled the following comments to help you prepare a revised submission.

The paper is in principle suitable for *eLife*, but the reviewers would like you to address the following comments. In addition to the original reviews (see below), Reviewer 2 added the following comment:

Was there any evidence for premature termination (in addition to pausing) at any genes? 7-8' retention of RNAPII is a stable pause, but how short-lived would RNAPII be at prematurely terminated genes? The authors see little evidence for this, but have HERV-H gene transcripts (which are reportedly high in ESCs and make lots of long noncoding RNAs) been checked and show any evidence of termination? Benkirane's Cell paper found that there is termination and Drosha-mediated cleavage at those genes (but in HeLa cells). Cramer also saw little evidence of premature termination in yeast in his Nrd1 paper, but HIV-1 LTR and HERV genes are thought to do this. The existing data should be checked for HERV-H genes, but obviously for the authors to reach their conclusions it is not essential to do this.

*Reviewer*
*1:*

This is a superb paper that should be published in *eLife*. In essence, the paper is simple, using GRO-seq to map Pol II on a genome-wide scale under conditions where transcriptional initiation or Pol II pausing are blocked by drugs (triptolide and flavopiridol, respectively). Of course, the validity of the study hinges on the specificity of the drugs, but both appear quite specific. Many conclusions of this paper have essentially been made previously; e.g., pausing as a general feature of Pol II transcription in mammals and other (but not all) organisms; stability of paused Pol II; gene-specific elongation rates; increased elongation speed at more distal position; effect of exons. However, what is particularly important about this paper is that it addresses all these fundamentally important issues in the most comprehensive, genome-wide form, and it doesn't use unusual inducing conditions. In addition, the use of the intronic RNA-seq data as a way of obtaining drug-independent data to support the conclusions is clever. Some of these important issues have been controversial or unclear in the literature, and the results here are as definitive as one could expect. So, while one could question whether the paper is truly novel, it represents a significant advance on a very important subject. The paper is generally well written, although some of the specific results and figures take some time to understand what has been really done.

1) The authors make the interesting observation that the decrease in Pol II association at downstream positions may be due to increased elongation rate as opposed to defective processivity due to low levels of premature termination. While I believe this conclusion for the physiological situations tested, the authors should mention that conditions that decrease elongation rate *can* cause the same effect; e.g., a slow Pol II, 6-AU, or inactivation of an elongation factor (37).

2) The term “transition point” needs to be defined. I got the general idea, but not the definition.

*Reviewer*
*2:*

This study provides a high-quality and quantitative assessment of transcription elongation rates in self-renewing mESCs from the Lis lab, inventors of the GRO-Seq technique. As generally is the case from this lab, the findings are interesting and conclusively supported, and should be of significant interest to the many investigators in this field. The main results provide strong support for numerous earlier studies arguing that elongation rates can vary significantly between genes, slow down through exonic regions, and correlates with CpG content and chromatin modifications like H3K79 methylation, while contradicting the idea that paused RNAPII complexes are rapidly terminated and replaced. The application of this technique to mouse stem cells is novel and well justified and will enrich existing datasets that for further studies seeking to correlate elongation rates with specific genic and epigenetic features.

One obvious limitation is the reliance on chemical inhibitors (FP and Trp) as surrogates for inactivation of Cdk9/P-TEFb and TFIIH, respectively. The authors should more fully acknowledge in the text that FP eliminates the expression of many short-lived proteins, including other transcription factors, which could affect these results, and that all of the conclusions are subject to potential off-targets effects of either drug. FP has a potent effect on histone modifications, not all of which have attributed to P-TEFb, and the authors should also address whether these changes would influence the elongation rates, which obviously had to be measured in the presence of the drugs.

In transformed cells FP induces cell cycle arrest and in some cases apoptosis – was there any effect on the cell cycle of these drugs in mESCs?

Kindly address how increased elongation rates correlate with increased H3K79me2 within long first exon, yet the highest elongation rates are measured at the 3' end of the genes, which are relatively lacking for H3K79me2?

The conclusion that paused RNAPII complexes are stable derives from their persistence for 7-8' after treatment with Trp, and leads the authors to conclude that premature termination is rare or undetectable in these cells – but it would be much more helpful to compare directly with a prematurely terminated gene – what did the paused complex look like at the highly active HERV-H element-containing, Drosha-sensitive, linkRNAs in these cells?

Because developmental lineage-specific genes in mESCs are not regulated by TFIIH, they may have distinct patterns of elongation, as may many signaling genes, and so the authors may want to stress that the 'universality' of their findings applies only to self-renewing ESCs.

---

## [Author Response]

*Was there any evidence for premature termination (in addition to pausing) at any genes? 7-8' retention of RNAPII is a stable pause, but how short-lived would RNAPII be at prematurely terminated genes? The authors see little evidence for this, but have HERV-H gene transcripts (which are reportedly high in ESCs and make lots of long noncoding RNAs) been checked and show any evidence of termination? Benkirane's Cell paper found that there is termination and Drosha-mediated cleavage at those genes (but in HeLa cells). Cramer also saw little evidence of premature termination in yeast in his Nrd1 paper, but HIV-1 LTR and HERV genes are thought to do this. The existing data should be checked for HERV-H genes, but obviously for the authors to reach their conclusions it is not essential to do this*.

We have addressed this interesting phenomenon in the mESC GRO-seq data we generated by looking at the specific genes found to have Drosha mediated cleavage and termination in HeLa cells by Wagschal et al. We have also looked at GRO-seq density profiles at HERV-H-like LTR repeats genome-wide to investigate the role of microprocessor and Xrn2 mediated termination within gene bodies in mESCs. The results of our additional analysis are discussed below in our response to the comments made by Reviewer 2.

Reviewer 1:

*1) The authors make the interesting observation that the decrease in Pol II association at downstream positions may be due to increased elongation rate as opposed to defective processivity due to low levels of premature termination. While I believe this conclusion for the physiological situations tested, the authors should mention that conditions that decrease elongation rate can cause the same effect; e.g., a slow Pol II, 6-AU, or inactivation of an elongation factor (*[37]*)*.

We agree that this careful study by Mason & Struhl should be acknowledged in this paper and have included references and a model comprising this work to the main text. We would like to point out that we looked at a model that fits the observations made by Mason & Struhl in our Monte Carlo simulation. The mixed model in Figure 4—figure supplement 1 takes into account parameters that predict that elongation rates and processivity are both low near the 5’ end of genes, while both processivity and elongation rate increase as Pol II progresses into the gene body. The elongation rate acceleration pattern and Pol II density profile that we predict from this model does not fit well with the actual measurements we made of the density profile and elongation rates throughout the gene body, indicating that this model is not the best description of genic transcription in normal cells. However, as the Mason & Struhl paper suggests, perturbation of either elongation rate or processivity by treating with 6-AU, or inactivation of an elongation factor could result in patterns of elongation rate and Pol II density as proposed by the mixed model.

Textual changes related to this comment: sentence beginning “The decreasing pattern of gene body Pol II density…” in the Results section entitled “Poll II stably elongates in the gene body, while termination is negligible”.

*2) The term “transition point” needs to be defined. I got the general idea, but not the definition*.

We apologize for not clearly defining what transition points are in the text. The transition point is the middle of the inhibition wave from the affected to unaffected region within the gene body after Trp or FP treatment. We changed the text to make this clear in the manuscript as well (Results section entitled “Measuring the speed of elongating Pol II in the gene body”:

Reviewer 2:

*One obvious limitation is the reliance on chemical inhibitors (FP and Trp) as surrogates for inactivation of Cdk9/P-TEFb and TFIIH, respectively. The authors should more fully acknowledge in the text that FP eliminates the expression of many short-lived proteins, including other transcription factors, which could affect these results, and that all of the conclusions are subject to potential off-targets effects of either drug. FP has a potent effect on histone modifications, not all of which have attributed to P-TEFb, and the authors should also address whether these changes would influence the elongation rates, which obviously had to be measured in the presence of the drugs*.

We agree with the reviewer that this study relies on the specificity of the drugs FP and Trp. Therefore, when we set out to do the GRO-seq experiments we ensured that we were using the minimal effective concentration of each of these drugs by doing western blots on the fractions of chromatin bound and free Pol II before and after drug addition (Figure 1—figure supplement 2), as well as ChIP-qPCR on the *ActB* gene to see what concentrations of FP and Trp clears Pol II from the gene body in a time dependent manner (not shown). We stress that both the concentration used for FP and Trp are at the lower spectrum of what is generally used for these drugs in the literature, reducing the risk of off-target effects. Furthermore, we use the drugs for short periods of time, 50 minutes is our longest timepoint, thereby minimizing secondary effects.

Even though we have taken considerable steps to perform the experiments at the minimal drug concentration possible, and measure within a relatively short period of time, we agree that the FP off-target effects may interfere with elongation, and therefore could hamper our elongation rate measurements. However, we would like to point out that we also measure elongation rates after Trp treatment. Trp blocks XPB helicase function, which is a vastly different mechanism of transcription inhibition compared to FP inhibition, and not likely to have similar potential off-target effects of FP. Although we could measure fewer transition points and elongation rates after treatment with Trp, the 683 transition points shared between the FP and Trp experiments did not differ significantly from one another, emphasizing that if there are off-target effects of FP that affect elongation rates, they are minor and within the error margins of our assay.

To address the reviewers concerns, we have made textual changes regarding the specificity of the drugs within the Results and Discussion sections:

*In transformed cells FP induces cell cycle arrest and in some cases apoptosis – was there any effect on the cell cycle*
*of these drugs in mESCs?*

Although we did not specifically look for the effects of FP or Trp on the cell cycle or cell death, we imagine that they are significant upon prolonged treatment with either drug, considering that both shut down virtually all Pol II transcription. However, within the 50 minutes of our assay, the ES cells were morphologically indistinguishable from the untreated samples. Moreover, other studies have used FP as a drug to block Cdk9 function for an hour or more, and at higher concentrations, without reporting major problems with the cells such as apoptosis.

We added a sentence to the Results section clarifying this (“Furthermore, we ensured that drug treated mESCs were morphologically indistinguishable from untreated cells”).

*Kindly address how increased elongation rates correlate with increased H3K79me2 within long first exon, yet the highest elongation rates are measured*
*at the 3' end of the genes, which are relatively lacking for H3K79me2?*

Based on our results we propose that both intrinsic and extrinsic factors determine Pol II elongation rate throughout the gene body. Intrinsic factors may consist of pausing factors that interact with the Pol II complex near the TSS and need to be removed or altered before the elongation complex reaches its full speed. This happens near the TSS, mostly within the first 15 kb of the gene body, based on the Pol II composite profiles and acceleration of Pol II within this region (Figure 4). Extrinsic factors comprise the genic features within the gene body that don’t directly interact with Pol II, but that do exert an effect on its elongation rate, such as histone modifications and exons (Figure 5).

To investigate what extrinsic features of DNA and chromatin could influence elongation rate we used the 938 mid elongation rates that were derived from the 12.5 to 25 min FP treatments. We used these elongation rates for two reasons: 1. This is the biggest group of elongation rates, providing the greatest statistical power, and 2. We only used elongation rates measured between these two timepoints for all genes to minimize the variation of intrinsic acceleration of Pol II in these genes. The average transition points after 12.5 or 25 min drug treatment are 7 and 30 kb respectively, which often coincides with the H3K79me2 enrichment with the first exon and intron. After exon 2, we saw a significant drop in H3K79me2 levels (Figure 5), but this is often outside the region were elongation rates are measured. Therefore, H3K79me2 has a positive effect on elongation rate early in the gene independently of the intrinsic acceleration. Downstream of the 25 min transition point, the intrinsic acceleration of Pol II has completed, and additional positive effects on elongation rate by H3K79me2 may no longer be needed.

To clarify this in the text we made the following changes: “…to increase statistical power, and to rule out major effects of intrinsic acceleration in the comparison between elongation rates. Existing genome-wide data of transcription factors, chromatin modifiers, and chromatin modifications were used to detect determinants of elongation rate (see for full list of features Figure 5—figure supplement 1)” was added to the Results section “Determinants of the elongation rates”.

“While CpG content is likely to exert effects throughout the gene, H3K79me2 is mostly enriched within exon and intron 1, indicating that this histone modification is beneficial to elongation early in the gene, while the elongation complex is still accelerating” was added to the last paragraph of the Results.

*The conclusion that paused RNAPII complexes are stable derives from their persistence for 7-8' after treatment with Trp, and leads the authors to conclude that premature termination is rare or undetectable in these cells – but it would be much more helpful to compare directly with a prematurely terminated*
*gene – what did the paused complex look like at the highly active HERV-H element-containing, Drosha-sensitive, linkRNAs in these cells?*

We agree and investigated the examples of genes and the proposed LTR dependent termination within gene bodies reported in the literature independently. The Wagschal et al. paper concludes two major points. First, microprocessor directly targets Pol II on HIV and HERV-H at their promoter region. Second, endogenous genes containing HERV-H elements in the gene body can also be affected by microprocessor targeting the elongating nascent RNA and causing premature termination in the gene body.

Although this study was carried out in human cells, it is possible that the same mechanism govern in the mouse ES cells. But currently, Drosha target sites have not been identified genome-wide in mouse ES-cells. This made it difficult for us to decide which sites to examine for premature termination. However, one study has previously shown that mouse endogenous retrovirus can trigger premature Pol II termination in the gene body (Jingfeng Li et al. 2012). Therefore, we made use of this information to answer two questions. First we asked whether the dynamics of promoter proximal Pol II is influenced on the potential Drosha target genes. Second is whether the Pol II termination is a general mechanism, taking place on sites with endogenous retrovirus sequences.

For the first question, we looked at the genes that are proposed to be regulated by termination in HeLa cells by Wagschal et al. (Yif1B, Tex14, Ptp4a1, Pde4B, Hhla1 (HERV-H) and Dlgap1). We found that 3 out of 6 genes were actively transcribed in mESCs, and calculated decay rates and half times for the promoter proximal Pol II in these genes (Figure 7). Each decay rate of the actively transcribed putative Drosha target genes was within the normal distribution of decay rates reported in Figure 2 (half times for Yif1B, Tex14, and Ptp4a1 were 6.6, 6.7 and 5.5 min, respectively), making a model in which pausing is affected by Microprocessor and Xrn2 dependent termination unlikely in these genes. Therefore, we do not find evidence of rapid rounds of Pol II termination at these promoters.

For the second question, we identified the IAP LTR endogenous retrovirus elements proposed to trigger Pol II termination in the Li et al. paper, and examined the average GRO-seq profile relative to these sites. If there were general Pol II termination occurring, we expect to see a drop of GRO-seq density downstream of these sites. We did not find such a drop significantly different from the alignment bias pattern (‘mappability’) of the regions, suggesting that Pol II termination is not generally occurring (Figure 7).

However, in the Polr1a gene discussed in the Li et al. paper, we observed the GRO-seq pattern indicative of premature nascent RNA cleavage and post-cleavage Pol II accumulation (Figure 7, black filled triangles) immediately downstream of the IAP LTR element (blue filled triangles). This is similar to the post poly-A accumulation of Pol II (black empty triangles). One possibility is that this Pol II accumulation originates from novel transcription initiation at the highly repeated region of the IAP LTR element. However, the fact that the accumulation of Pol II at this region is not resolved by FP treatment shows that the initiation site is distant, likely the promoter of the Polr1a gene, not a site inside the IAP LTR element. Although the precise mechanism cannot be determined for now, we believe this could be a case of microprocessor dependent Pol II termination specifically happening at this site.

Overall, we conclude that premature termination in the gene body is not a general mechanism, but rather an exceptional case depending on sequence specific sites that can recruit microprocessor complex to the nascent RNA. Because this one possible case of termination that we uncovered does not change our conclusions, we prefer not to address this complicated site-directed phenomenon in this already lengthy manuscript.Author response image 1.LTR.A. The decay rate plots of the three active genes identified by Wagschal et al. to be regulated at the level of termination of Pol II at the promoter or within the gene body in Hela cells. B. The composite profile of GRO-seq density around intragenic LTR sites genome wide in blue, compared to the mappability of these sites in red. C. The GRO-seq density profile before and after FP treatment and the RNA-seq profile of the LTR containing gene Polr1a shows a potential example of genic termination mediated by the IAP LTR (blue rectangle) recruited microprocessor and Xrn2. The black rectangle indicates the GRO-seq pattern consistent with premature nascent RNA cleavage and post-cleavage Pol II accumulation, while the empty triangle indicates the post poly-A accumulation of Pol II.

*Because developmental lineage-specific genes in mESCs are not regulated by TFIIH, they may have distinct patterns of elongation, as may many signaling genes, and so the authors may want to stress that the 'universality' of their findings applies only to self-renewing ESCs*.

The paper that discusses the role of TFIIH in the transcription of developmental genes in mESCs by the Reinberg lab has very recently been published, so it only just has come to our attention. In this paper, Tee et al. find, among other interesting results, that Erk2 kinase activity replaces TFIIH function at a number of developmental lineage genes, as tested by looking in mESCs at the effects of 30 min of 1 µM Trp treatment on developmental and mESC specific genes with ChIP-qPCR on the Ser5 phosphorylated CTD of Pol II. Interestingly, developmental genes were unaffected by Trp treatment, indicating a TFIIH independent initiation mechanism. We were unable to find the datasets generated for this paper (the reported SRA accession number SPR028688 did not retrieve any data in various databases), making it difficult to assess the Erk kinase, Ercc3 and Ser5 and total Pol II ChIP-seq data ourselves. Therefore, we used the Ser5, Ser2 and N-terminal Pol II ChIP-seq datasets generated by Rahl et al. (Cell, 2010), as well as our own GRO-seq datasets, before and after 25 min Trp treatment to best mimic the 30 min Trp treatment used by Tee et al., to look at the patterns of expression of the genes insensitive to Trp. Surprisingly, we could not reproduce the results reported by Tee et al. All actively transcribed genes reported as being insensitive to Trp treatment by Tee et al. were clearly inhibited by 25 min Trp treatment in our study, or not actively transcribed (Figure 8 and data not shown).

To understand the discrepancy between the two studies, we considered a number of differences between the Tee et al. experiments and our own. Firstly, the genetic background and growth conditions for the cell lines used in both papers may not be identical (we used V6.5 whereas the mESCs strain used in Tee et al. was not clearly stated). Thus, strain and growth condition differences could result in variation in gene expression. Indicative of this potential difference in gene expression, in our study and that of the Young lab, the Trp insensitive developmental genes were far less transcribed in V6.5 mESCs compared to the control genes Klf2 and Trim28 (transcribed ∼10 x higher as measured by ChIP- and GRO-seq, note the y-axes for each browser shot). In contrast, the level of transcription for these genes as measured by Pol II-Ser5 ChIP-qPCR in Figure 5 of Tee et al. appears within the same order of magnitude for both control and Trp insensitive developmental genes, indicating that these genes are similarly active in the mESCs used in their experiment.

Alternatively, the antibody used in the Tee et al. study may target a different form of Pol II than detected in the GRO- and ChIP-seq data shown in the figure describing TFIIH sensitivity (Figure 8, below). This also is suggested by the Pol II-Ser5 ChIP-seq pattern of the browser shots shown in Figure 5 of the Tee et al. paper. The Ser5-Pol II signal looks strikingly different than total Pol II patterns, while these forms of Pol II tend to overlap in other studies, and with GRO-seq patterns (Figure 8, ([47]; Hintermair et al. 2012)), indicating that there may be an alternative form of Pol II detected in the Tee et al. paper. Regardless of the form of Pol II detected by Tee et al., the active and engaged Pol II that is detected by GRO-seq at these poorly transcribed developmental genes is susceptible to TFIIH inhibition and block of initiation by 25 min Trp treatment in V6.5 mESCs.Author response image 2.TFIIH sensitivity.The genes assayed by ChIP-qPCR of Ser5 phosphorylated Pol II in Figure 5 of Tee et al. Cell 2014 were examined. In the top box, two Trp sensitive control genes taken from this figure are shown, and in the bottom box, four examples of Trp “insensitive” genes are displayed. The tracks shown are Ser2 (green), Ser5 (purple) and N-terminal Pol II (black) ChIP-seq datasets from [47] Cell. The y-axis in RPKM is identical for the three ChIP-seq tracks. Below, the control DMSO and 25 min Trp treated GRO-seq datasets are displayed, with the sense strand in red and the antisense strand in blue. The y-axis for the GRO-seq data is in red and blue also and identical for both GRO-seq datasets, but different from the ChIP-seq y-axis due to different normalization done with Spike-in control instead of total reads as with ChIP-seq. Location of the genes is shown in red or blue at the bottom of each screenshot, and the location of the qPCR amplification product, as reported by Tee et al., is shown in green above the gene. Note that the Trp sensitive control genes Klf2 and Trim28 are over ∼10x higher expressed than the Trp “insensitive” developmental genes. The expression as measured by both GRO- and ChIP-seq of the HoxC5 and T genes was negligible. Despite the low activity of the developmental genes Gata4 and Fgf5, a Trp dependent block of initiation and loss of transcription after 25 min Trp treatment could clearly be detected.

Because of the clear differences in results between these two manuscripts, we are uncomfortable including a sentence regarding TFIIH insensitive genes in mouse ES cells or other cell types. We would need to address this issue in depth to make a sensible statement, including analyzing the Tee et al. data sets that are currently not available, as well as doing experiments where we combine Trp treatment with differentiation. However, we feel that this is beyond the scope of our current paper.